evolution/computational biology

sign language, language phylogeny, language evolution, phylogenetic networks

**Author for correspondence:**
Justin M. Power
e-mail: justin.power@utexas.edu

# Evolutionary dynamics in the dispersal of sign languages

Justin M. Power[1], Guido W. Grimm[2]

and Johann-Mattis List[3]

[1]Department of Linguistics, University of Texas at Austin, Austin, TX, USA
[2]Independent researcher, Orléans, France
[3]Department of Linguistic and Cultural Evolution, Max Planck Institute for the Science of Human History, Jena, Germany

 JMP, 0000-0001-5695-5357; GWG, 0000-0003-0674-3553;
J-ML, 0000-0003-2133-8919

The evolution of spoken languages has been studied since the mid-nineteenth century using traditional historical comparative methods and, more recently, computational phylogenetic methods. By contrast, evolutionary processes resulting in the diversity of contemporary sign languages (SLs) have received much less attention, and scholars have been largely unsuccessful in grouping SLs into monophyletic language families using traditional methods. To date, no published studies have attempted to use language data to infer relationships among SLs on a large scale. Here, we report the results of a phylogenetic analysis of 40 contemporary and 36 historical SL manual alphabets coded for morphological similarity. Our results support grouping SLs in the sample into six main European lineages, with three larger groups of Austrian, British and French origin, as well as three smaller groups centring around Russian, Spanish and Swedish. The British and Swedish lineages support current knowledge of relationships among SLs based on extra-linguistic historical sources. With respect to other lineages, our results diverge from current hypotheses by indicating (i) independent evolution of Austrian, French and Spanish from Spanish sources; (ii) an internal Danish subgroup within the Austrian lineage; and (iii) evolution of Russian from Austrian sources.

## 1. Introduction

The human capacity for language is not limited to the oral–aural modality. Instead, *homo symbolicus* has also developed complex natural language in the gestural–visual modality, particularly in deaf signing communities throughout the world. Signing communities have likely existed throughout human history in communities with high incidences of deafness [1,2] and in urban areas [3,4], though few records survive of such communities or their languages prior to the eighteenth century.

The development of educational institutions for the deaf, which began during the Enlightenment in Europe in the late eighteenth and early nineteenth centuries, contributed to the formation of stable signing communities and the emergence of widespread, conventional sign languages (SLs) [5,6]. In these institutions, relatively large numbers of deaf children were exposed to accessible language at early ages, and signed language was transmitted naturally from older to younger signers [7]. *Manual alphabets* (MAs)—forms representing an alphabet, in which one form corresponds to one letter—also came into more widespread use in this period. While the exact origins of the MAs used in European educational institutions are unclear, scholars point to Melchor de Yebra's *Refugium Infirmorum*, published in Madrid in 1593, and to *Digiti Lingua*, published anonymously in London in 1698, as the earliest records of one- and two-handed MAs that are the likely progenitors of many contemporary MAs [8,9].

The languages and MAs that emerged in these newly formed signing communities were soon dispersed to other parts of Europe and beyond. For example, the success of the first public school for the deaf, the *Institut National de Jeunes Sourds de Paris*, founded between 1759 and 1771 [8], attracted educators from across Europe and the Americas, who came to the Paris Institute to learn pedagogical methods, with the goal of establishing schools for the deaf in their countries of origin [10]. Deaf students from other countries also came to Paris, graduated and returned home to work as teachers or to found new schools [11,12]. Thus, linguistic connections were formed between geographically distinct signing communities. Many of these connections have been reported previously in multiple sources [7,13]. The spread of French SL via educators and educational institutions has been well documented, including connections within Europe, such as The Netherlands in 1790 [14]. In addition, French educators helped establish schools for the deaf in the Americas, including the USA in 1817 [15], Brazil in 1857 [16] and Mexico in 1867 [17]. Based on these institutional links and, in some cases, on small-scale linguistic studies [18], scholars have posited a French SL family [19]. Other well-known SL dispersals include Swedish SL to Portugal in 1823 [20]; Danish SL to Norway in 1825 [21] and Iceland in 1867 [22]; and British SL, with its two-handed MA, to Australia in 1825 and New Zealand in 1868 [23] (see electronic supplementary material, 2.3, table S3 for a detailed overview of reported connections among SLs). These historical connections may form the basis of other putative language families, as has been suggested in the case of the so-called BANZSL family (British, Australian and New Zealand SLs) [24].

However, the histories of many other SLs are less clear. For example, founders of the first school for the deaf in Vienna, Joseph May and Friedrich Storch, visited the Paris Institute in 1777 to learn pedagogical methods and to subsequently establish deaf education in the Habsburg Empire [25]. Perhaps because of these historical connections, it has been thought that Austrian SL is related to French SL, sharing a common ancestor [19]. However, this connection appears to be based only on inferences from the institutional connection, as there have been no published comparisons of the languages themselves. Similarly, Peter Atke Castberg, who founded the first school for the deaf in Copenhagen in 1807, visited deaf educational institutions in Germany, France and Austria between 1802 and 1805 [20], but it is not known how Danish SL relates to these other SLs. While Austrian- and French-trained educators helped establish the first school for the deaf in Russia in the period from 1806 to 1810 [26,27], little is known about whether Austrian and French SLs may have influenced Russian SL linguistically [28]. Moreover, multiple sources have been reported for several SLs, such as Austrian, French, German and Russian influence on Polish SL [29,30], but little is known about the putative linguistic contributions of each source. International Sign, which became partly standardized in the second half of the twentieth century to facilitate communication at conferences of the World Federation of the Deaf [31], may have had a homogenizing effect on the SLs of member countries, some of which adopted the newly formed International Sign MA [32].

Thus, while the history of deaf educational institutions has been well documented in many countries, much less is known about how SLs themselves relate to one another. Whereas the world's spoken languages have been classified in families and subfamilies based on their evolutionary histories, few attempts have been made to form large-scale genetic classifications of the world's SLs [19,33], which are typically missing from overviews of the world's language families (e.g. [34]; see [35,36] for overviews including SLs). To our knowledge, the only published large-scale evolutionary history of SL families based at least partly on language data is Anderson [19], which shows two separate trees dividing 'South-West European' from 'North-West European' SLs. Contemporary SLs in the 'south-west' language family are subgrouped in three main branches, corresponding to French, Polish and Spanish lineages; while the 'north-west' language family is divided into British, German and Swedish lineages. Unfortunately, little information is provided about the methods used to identify relationships among

these SLs or for how the families were subgrouped. Thus, despite Anderson's pioneering overview of potential relationships among SLs, a lacuna remains in our knowledge for much of the history of an entire class of human languages. The difficulty in forming classifications of SLs is due in part to challenges in understanding the evolutionary processes that have shaped the diversity of contemporary SLs. In particular, using traditional historical comparative methods, sign researchers have been unable to distinguish the results of tree-compatible evolutionary processes—that is, patterns of similarity reflecting inheritance in a vertical ancestor–descendant relationship—from tree-incompatible processes, such as borrowing and convergence. As a consequence of these methodological challenges, comparative studies of SLs at times conflate vertical and horizontal relationships in forming SL families [37] or these studies forgo historical interpretations of their results [38].

While relationships among spoken languages have been studied using both traditional methods and, more recently, computational phylogenetic methods [39,40], to date no published studies have attempted to use phylogenetic methods to infer relationships among SLs on a large scale. As a first step in investigating the evolutionary histories of SLs and the processes that have shaped them, we use network-based *exploratory data analysis* (EDA) [41] with a sample of 76 SL MAs (40 extant and 36 historical). Our approach makes use of *data-display networks*, which represent both tree-compatible and tree-incompatible patterns within a dataset and are therefore a useful starting point for understanding evolutionary processes and formulating phylogenetic hypotheses.

# 2. Material and methods

## 2.1. Data

We created a dataset of 76 MAs, comprising 2124 total entries from contemporary (40; electronic supplementary material, 2.1, table S1) and historical (36; electronic supplementary material, 2.2, table S2) sources, both print and online, the geographical distribution of which is illustrated in figure 1. Current estimates of the total number of SLs in the world range between 144 [35] and 193 [36]. However, easily accessible MA data are available for relatively few of these. SLs included in this study are those for which we were able to find quality MA data either freely available online or in print sources published in the respective countries. We transcribed MA handshape forms using HamNoSys [42], a transcription system for SLs. HamNoSys is designed to transcribe the four major phonological parameters of the sign—handshape, orientation, location and movement—and includes symbols for non-manual features, as well as types of symmetry in two-handed signs. We coded each MA form for morphological similarity using criteria described in more detail below. We used the EDICTOR tool [43], which was originally developed for the curation and analysis of historical comparative spoken language data, to store the dataset, our manually established similarity judgements and our analyses in a transparent way that can be easily inspected and edited by researchers, while at the same time being accessible for machines (electronic supplementary material, S1). In addition, we share the data in the TSV format required for processing by the LingPy software package [44], and in CLDF format, following the recommendations of the Cross-Linguistic Data Formats Initiative [45], in order to ensure that the data can be easily reused, checked for errors and potentially improved upon by colleagues.

Historical comparative studies of languages typically use basic vocabulary as comparanda, but there are good reasons to begin with a comparison of MAs instead. First, many of the historical contexts in which MAs were created and transmitted have been well documented, as have the MAs used in these communities. There exist far fewer historical dictionaries of the world's SLs compared with historical examples of MAs, though some do exist [46–48]. Thus, the data provide a relatively well-understood test case for studying evolutionary processes in SLs. Second, while computer-readable transcription systems have been developed for SLs—HamNoSys [42] and SignWriting [49] are the most commonly used—these are still not widely used in SL lexicography (see [50] for a corpus-based dictionary including transcriptions in HamNoSys; see [51] for a dictionary with transcriptions in SignWriting), and transcriptions in the two systems are not straightforwardly comparable in all respects. Open, computer-readable, cross-linguistic comparative datasets of SL vocabulary do not yet exist, due in part to the lack of consensus on transcription system, but also to the time-consuming nature of SL transcription. Transcribing handshapes in MAs instead of lexical signs—which typically include specifications for orientation, location and movement in addition to handshape—significantly reduces the time necessary to create a large cross-linguistic comparative dataset.

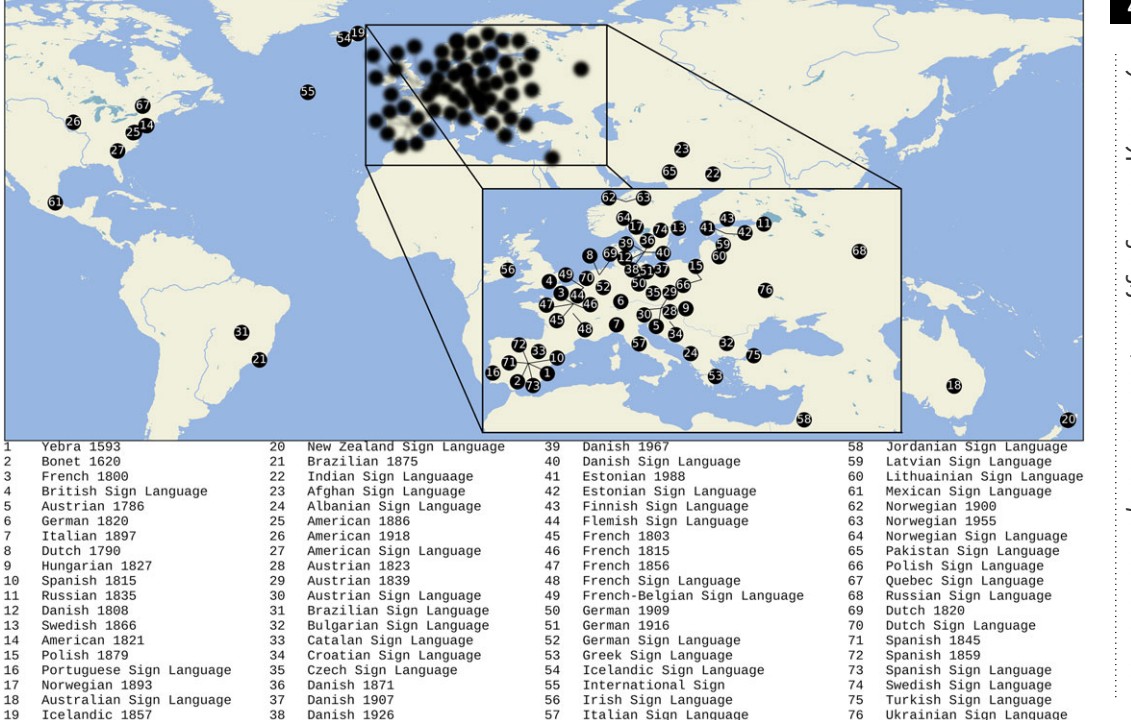

| 1 | Yebra 1593 | 20 | New Zealand Sign Language | 39 | Danish 1967 | 58 | Jordanian Sign Language |
|---|---|---|---|---|---|---|---|
| 2 | Bonet 1620 | 21 | Brazilian 1875 | 40 | Danish Sign Language | 59 | Latvian Sign Language |
| 3 | French 1800 | 22 | Indian Sign Languaage | 41 | Estonian 1988 | 60 | Lithuainian Sign Language |
| 4 | British Sign Language | 23 | Afghan Sign Language | 42 | Estonian Sign Language | 61 | Mexican Sign Language |
| 5 | Austrian 1786 | 24 | Albanian Sign Language | 43 | Finnish Sign Language | 62 | Norwegian 1900 |
| 6 | German 1820 | 25 | American 1886 | 44 | Flemish Sign Language | 63 | Norwegian 1955 |
| 7 | Italian 1897 | 26 | American 1918 | 45 | French 1803 | 64 | Norwegian Sign Language |
| 8 | Dutch 1790 | 27 | American Sign Language | 46 | French 1815 | 65 | Pakistan Sign Language |
| 9 | Hungarian 1827 | 28 | Austrian 1823 | 47 | French 1856 | 66 | Polish Sign Language |
| 10 | Spanish 1815 | 29 | Austrian 1839 | 48 | French Sign Language | 67 | Quebec Sign Language |
| 11 | Russian 1835 | 30 | Austrian Sign Language | 49 | French-Belgian Sign Language | 68 | Russian Sign Language |
| 12 | Danish 1808 | 31 | Brazilian Sign Language | 50 | German 1909 | 69 | Dutch 1820 |
| 13 | Swedish 1866 | 32 | Bulgarian Sign Language | 51 | German 1916 | 70 | Dutch Sign Language |
| 14 | American 1821 | 33 | Catalan Sign Language | 52 | German Sign Language | 71 | Spanish 1845 |
| 15 | Polish 1879 | 34 | Croatian Sign Language | 53 | Greek Sign Language | 72 | Spanish 1859 |
| 16 | Portuguese Sign Language | 35 | Czech Sign Language | 54 | Icelandic Sign Language | 73 | Spanish Sign Language |
| 17 | Norwegian 1893 | 36 | Danish 1871 | 55 | International Sign | 74 | Swedish Sign Language |
| 18 | Australian Sign Language | 37 | Danish 1907 | 56 | Irish Sign Language | 75 | Turkish Sign Language |
| 19 | Icelandic 1857 | 38 | Danish 1926 | 57 | Italian Sign Language | 76 | Ukrainian Sign Language |

**Figure 1.** Contemporary and historical SLs in our sample, with locations being derived from Glottolog [36] for contemporary languages and from city of publication for historical MAs.

However, while using MA data has facilitated the creation of a cross-linguistic comparative dataset for this study, there are important differences between an historical comparative study of basic vocabulary and our study focusing on MAs, differences that may affect inferences about the histories of the languages in our sample. Compared with basic vocabulary, an MA may be less integrated within a language or used less frequently. For example, some handshapes and handshape features may only be found in an SL's MA [52] or in a limited subset of an SL's lexicon, as in *initialized* signs, which incorporate in a lexical sign the MA handshape representing the initial letter of a direct translation from a written language [53]. In addition, there is variation both within and across signing communities in the frequency of usage of *fingerspelling*, the representation of a written word with a sequence of MA handshapes. In some SLs, fingerspelling is used frequently in everyday discourse [54], and young children acquire MA forms and fingerspelling patterns, along with lexical signs, before they are able to read [55,56]. In other SLs, fingerspelling may be used less frequently [9], and frequency of usage may vary across signers within the same signing community [57]. An MA that is less integrated within an SL may, in consequence, be more readily subject to replacement compared with basic vocabulary. The replacement of MAs by the International Sign MA indeed happened in several of the Austrian-origin SLs, as the results in §3 show. Thus, sampling from MAs instead of basic vocabulary may affect historical inferences about the evolutionary histories of SLs. However, we show here that many of the historically attested extra-linguistic connections among SLs are represented clearly in the network analyses in §3. We take this as confirmation that coherent, historically relevant information is recoverable from a dataset consisting of MAs.

## 2.2. Character coding

Each MA form includes a specification for handshape with a particular spatial orientation and may include a characteristic location and movement. For determining similarity of MA forms, we considered only handshapes and movements, as these are represented most consistently in both contemporary and historical sources. In many historical sources, it is not possible to determine actual spatial orientations and locations because handshapes are depicted without reference to the body. For this reason, we did not consider orientation or location in determining the similarity of MA forms. We judged MA forms to be similar based on specifications for finger extension, finger bending,

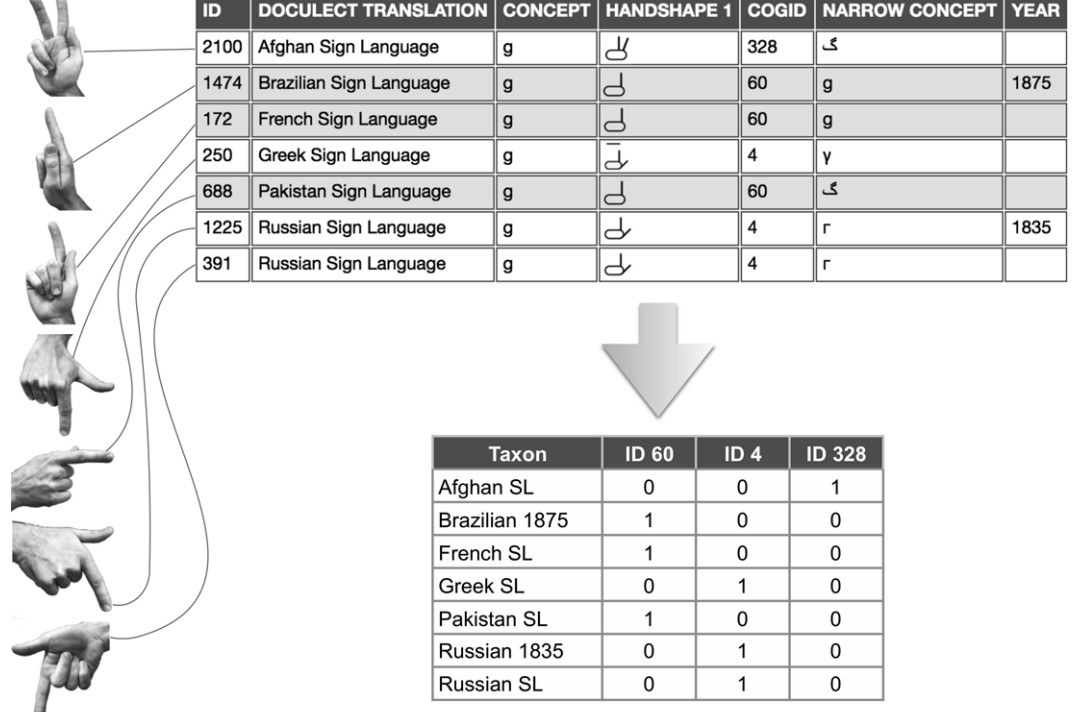

**Figure 2.** Simplified coding example for handshapes representing Latin ⟨g⟩, and its counterparts in Cyrillic, Greek and Persian/Urdu.

extended finger separation, thumb configuration and movement. If these specifications were all similar, we coded MA forms with the same ID, as described below in our explanation of figure 2. If any of these specifications differed for two MA forms, we coded the forms with different IDs. In coding morphological similarity, judgements must be made about non-discrete characters with potentially infinitely fine-grained differences. This is particularly challenging in judging similarity in finger bending, but is relevant to all aspects of the comparison. We have made the data and our similarity judgements openly accessible to researchers for testing other methods for determining similarity (see electronic supplementary material, S1).[1]

In coding similarity across MAs representing different types of alphabets—for example, Latin and Cyrillic alphabets—it is possible to consider the *form* of the grapheme, or the *sound* represented by the grapheme as bases for organizing the comparison. For example, the graphemic forms of Latin ⟨b⟩ and Cyrillic ⟨В⟩ are similar, but they represent different sounds: the voiced bilabial stop (IPA [b]) for Latin and the voiced labiodental fricative (IPA [v]) for Cyrillic. Thus, a decision must be made about whether to organize the comparison based on (i) form, by comparing handshapes representing Cyrillic ⟨В⟩ with handshapes for Latin ⟨b⟩; or instead, based on the (ii) sound represented, by comparing handshapes for Cyrillic ⟨В⟩ with those for Latin ⟨v⟩. In many such cross-alphabet comparisons, the documentary record is suggestive about how the comparisons should be organized. In the example just mentioned, we decided to compare handshapes representing Cyrillic ⟨В⟩ with Latin ⟨b⟩ and not ⟨v⟩, thus opting for variant (i), because historical records provide clues about how the Russian SL MA was adapted from other European MAs in the early nineteenth century [27]. When graphemic forms across two alphabet types are similar, as in the case of Latin and Cyrillic just mentioned, we took the form of the grapheme as the basis of comparison. However, there are many graphemes that differ in form across Latin-based, Cyrillic-based, Greek and Arabic-based alphabets, all of which are represented by MAs in our sample. In such cases, the sound represented by the grapheme is the only basis for comparison. The technique of parsimony character mapping on networks (see electronic supplementary material, 4.2, figures S2–S4) allows identification of concepts that support a split in the network. The technique helps to indicate how the coding of individual concepts contributes to the overall differentiation pattern. As this pattern may be masked by the

---

[1]JMP coded the similarity judgements. As with cognate judgements in historical comparative research on spoken languages, decisions may vary across coders. An alternative approach is to agglomerate the judgements of multiple coders, but this would have gone beyond the scope of the exploratory study presented here.

aggregated distances upon which the network was reconstructed, character mapping can make it possible to render the underlying data of a network transparent.

In figure 2, we exemplify our coding approach and the resulting binary matrices for use in phylogenetic methods. For reasons of space, we limit the number of taxa in the example to those necessary for detailing our methods, and provide a second, more complex example in the electronic supplementary material, 3.2, figure S1. The value in the 'Concept' column—'g' in this example— enables comparisons across alphabets. The left side of figure 2 depicts handshape forms representing four different graphemes, three of which represent the voiced velar stop (IPA [g]): Latin ⟨g⟩, Cyrillic ⟨Г⟩ and Persian/Urdu ⟨گ⟩; and one for the voiced velar fricative (IPA [γ]): Greek ⟨γ⟩. Starting at the top of the figure, Afghan SL represents the grapheme ⟨گ⟩ by extending the index and middle fingers, a form that is unique in this comparison. To represent the grapheme ⟨g⟩, historical Brazilian SL (Brazilian 1875) and French SL use forms with an extended index finger and the thumb orientated in a similar direction. Pakistan SL uses a similar handshape for the grapheme ⟨گ⟩. Finally, Greek SL, Russian SL and historical Russian SL (Russian 1835) represent ⟨γ⟩ and ⟨Г⟩ with extended index finger and thumb extended outward.

In the upper right side of figure 2, we show how these handshapes were coded in the EDICTOR tool [43]. Morphological similarity was coded by assigning the same numerical identifier (an arbitrary number per cognate set) in the 'CogID' (= Cognate Identifier) column for languages with similar forms. The column 'Narrow concept' tracks the grapheme represented in each MA, and 'Year' indicates when historical sources were published, an empty cell in the 'Year' column reflecting that the source is contemporary. Afghan SL, not being similar with any other taxon, was assigned ID 328. We coded the handshapes in Brazilian 1875, French SL and Pakistan SL as similar and assigned them ID 60. Handshapes in Greek SL, Russian SL and Russian 1835 were coded as similar and assigned ID 4. The bottom of figure 2 shows how our coding translates to the binary matrix for use in phylogenetic methods. Taking the character ID from the 'CogID' column, Brazilian 1875, French SL and Pakistan SL were scored as 1 for ID 60, while the other four taxa were scored as 0. Greek SL, Russian 1835 and Russian SL were scored as 1 for ID 4, and the other taxa 0. Afghan SL was scored as 1 for ID 328, while the other taxa were scored as 0.

## 2.3. Phylogenetic analysis

The matrix includes characters supporting language cliques that are compatible (inherited patterns) and incompatible (horizontally propagated patterns) with the splits in the unknown true tree. The true tree is, in this case, not necessarily a sequence of dichotomous splits because the tree can be polytomous and anastomizing: (i) one ancestor may have more than two direct descendants; and (ii) a descendant may have more than one direct ancestor in the case that, for example, an MA is the product of combining two (or more) different sources. Thus, any simple, dichotomizing tree that we infer or select using tree-reconstruction methods and commonly used optimality criteria will be incomprehensive, and any signal in the matrix reflecting aspects not covered by the tree's selected or inferred topology will add to data incompatibility (see electronic supplementary material, 4.1). Another source of tree-incompatible signal is the inclusion of putative 'ancestors' in the form of historical MAs, as well as their direct or distant 'descendants'. Spencer *et al.* [58] showed that the distance-based Neighbour-Nets (NNets; [59]), which were designed to counter the problem of signal incompatibility, outperform tree inferences when it comes to correctly depicting ancestor–descendant relationships. NNets were inferred for the complete taxon set and time-filtered taxon subsets: all MAs from before 1840, later historical MAs including youngest pre-1840 MAs as reference points (potential sources) and contemporary and post-1950 MAs.

With respect to the complex signal in the underlying matrices, we thus relied exclusively on network-based EDA [41,60] using planar (two-dimensional), distance-based (NNets) and multidimensional, tree-sample-based splits graphs (*Support Consensus Networks* [61,62], CNets). While NNets handle signal incompatibility well, they are restricted to two dimensions. In our case, an MA may derive from more than two sources and can show affinities to more than two unrelated groups of SLs. Such complex relationships may be captured in a bootstrap pseudoreplicate sample and visualized using CNets (see electronic supplementary material, 4.3, figures S5 and S6A–D). For instance, if an MA *xy* shares CogIDs with three different SL groups, the bootstrap replicate trees may reflect this situation by showing a three-way split support for each topological alternative (*xy* is placed within a different group in each pseudoreplicate tree) and an according three-dimensional box in the CNet. We used PAUP* [63] to compute simple (Hamming) pairwise distances as the basis for the NNets and establish

non-parametric bootstrapping (BS) branch support under the Least-Squares (LS) and Maximum Parsimony (MP) optimality criteria. BS analysis used 10 000 pseudo-replicates; replicate trees were inferred using the BioNJ algorithm [64] for LS and quick-and-dirty BS for MP as outlined by Müller [65] ('MulTrees' option deactivated; only one tree saved per replicate). For Maximum Likelihood (ML) BS support, we used 10 000 replicate trees generated with RAxML v.8.0.20 [66] and the standard model for binary data allowing for site variation modelled via the Gamma function, and corrected for ascertainment bias (recommended setting for binary data without invariable sites; the effect has only been tested for phylogenomic binary data; hence, we also ran the same analysis without correcting for ascertainment bias). Splits graphs were inferred with SplitsTree v.4.13.1 [67], either using the distance matrix or BS pseudoreplicate tree sample as input. NEXUS-formatted matrices and Splits-NEXUS-formatted splits graphs are included in the electronic supplementary material (Folders 'Matrix', 'Networks').

Post-analysis character mapping was done by hand-and-eye following the logical framework of median networks and guidelines provided by Bandelt *et al.* [68] for their manual reconstruction. In contrast to a dichotomized and/or anastomized tree, a median network considers taxa to be either tips or medians, representing ancestral variants connecting the tips and can be multi-dimensional. A full median network includes all possible most-parsimonious solutions for the mutation of a character, character complex or data matrix. For this study, we establish the minimum amount of necessary changes in each set of binary sequences representing concepts found in the standard Latin alphabet (letters ⟨a⟩ to ⟨z⟩) along time-filtered networks (figures in electronic supplementary material, 4.2 and file 'lists.xlsx').

## 3. Results

The NNet in figure 3 allows defining eight main groups of differing coherence and uniqueness. Each group forms a neighbourhood in the graph defined by a single, more or less prominent, edge-bundle. Two of the groups collect SLs of (i) Austrian- and (ii) French-origin; the oldest SLs in these groups (Austrian 1786, French 1800) may reflect the common bases from which the SLs in these groups are derived. The largely extinct Austrian-origin group includes a single surviving contemporary SL, Icelandic SL. Most other contemporary SLs (e.g. Austrian, Danish and German SLs) of the Austrian-origin group are now found in the French-origin group, which includes the International Sign MA. In addition, we recognize (iii) a British-origin group; (iv) an Afghan-Jordanian group, with lowest overall dissimilarity to the British-origin group; (v) a Polish group that is connected with (vi) the Russian group via Latvian SL; (vii) a distinct Spanish group including the oldest MAs in our dataset (Yebra 1593, Bonet 1620) and (viii) the very unique Swedish group which includes Portuguese SL. The spiderweb structure of the centre of the NNet graph indicates that the data cannot resolve the principal relationships between the distinguished eight main groups.

The robustness of discriminating signal in the underlying matrix for each main group, estimated using non-parametric BS support, is shown in table 1. In general, highly coherent, distinct groups (Afghan-Jordanian, British-origin, Polish, Spanish, Swedish groups) received moderate to high support (BS > 48; usually more than 90 for at least two optimality criteria; electronic supplementary material, 4.3, see also file 'lists.xlsx') irrespective of the optimality criterion used. Less coherent groups (Austrian-origin, French-origin, Russian groups) received low support (BS < 42). This demonstrates that the distance matrix well reflects the overall diversity patterns. Inter-group relationships are essentially unresolved: best-supported alternatives have a BS ≤ 23. Ambiguous BS support (i.e. BS ≪ 100) can result from a lack of discriminating signal or internal signal conflict, which can be explored using CNets (see electronic supplementary material, 4.3). In the case of the low-supported Austrian- and French-origin groups, no alternative finds a BS ≥ 15; these groups are poorly supported but lack alternatives. The same holds for the much higher BS support of the Spanish group. In the case of the Russian group, the low support relates to competing alternatives: the data prefer and would support partly incongruent tree-topologies (electronic supplementary material, 4.3). The two major sources of signal conflict are (i) Latvian SL, which is substantially less dissimilar to the Polish group than all SLs of the Russian group ($BS_{NJ} = 37$, but $BS_{ML,MP} < 15$), hence, its intermediate placement in the NNet (figure 3) and (ii) the Russian 1835 MA. In this case, the BS support values can vary substantially between optimality criteria: the distance-based NJ versus the character-based, mutation-probability naive MP versus the character- and model-based ML. Within the Russian group, Russian 1835 is most closely related to contemporary Cyrillic-representing MAs, while differing from Estonian SL, Estonian

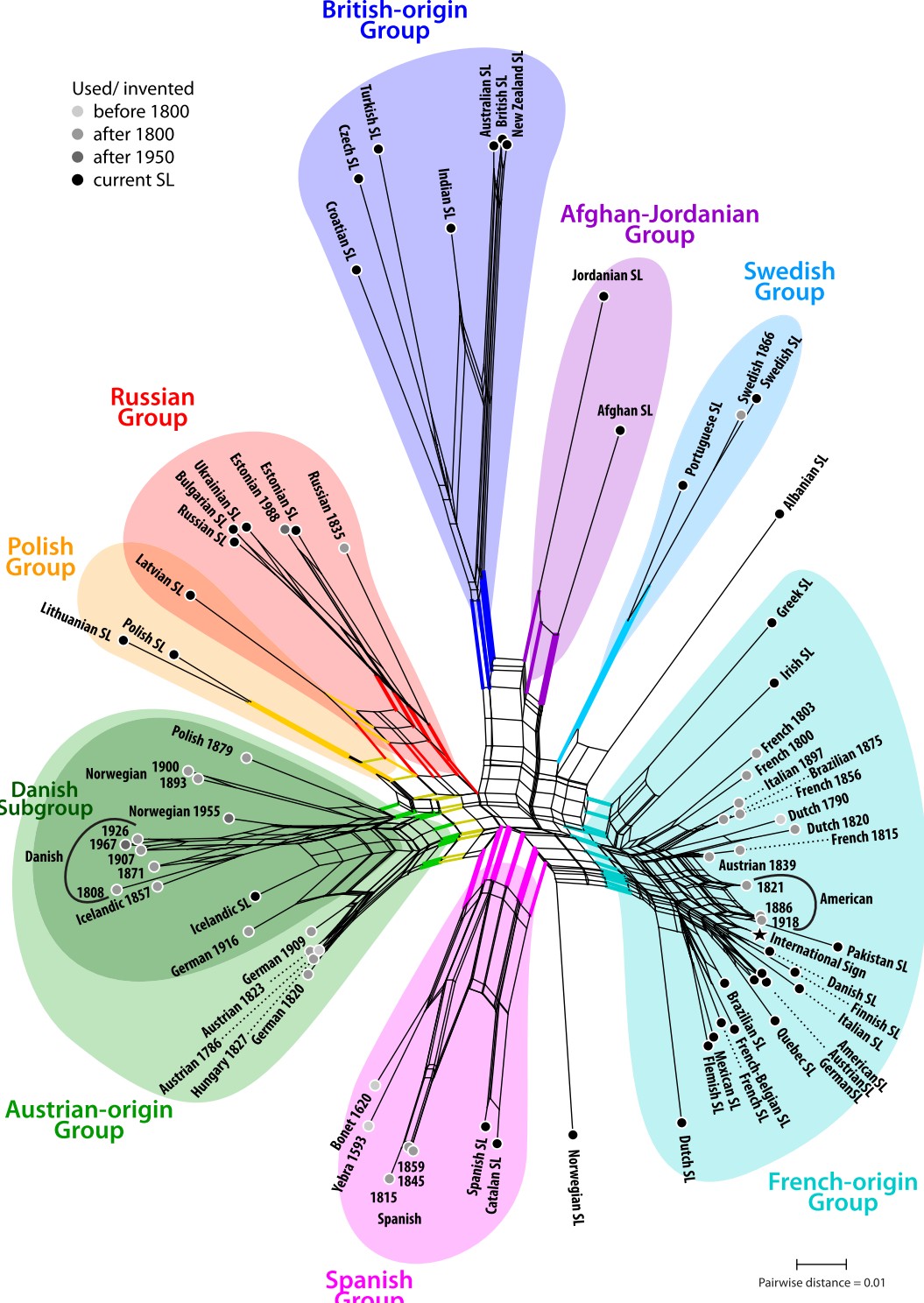

**Figure 3.** Neighbour-net based on simple (Hamming) pairwise distances calculated from the standard-coded CogID binary matrix. Colours highlight the main groups and the Danish subgroup (cf. figure 4; electronic supplementary material, 4.2, figures S2–S4) within the Austrian-origin lineage. Neighbourhood-defining edge-bundles are also highlighted.

1988 and Latvian SL. In the planar NNet, this conflict is resolved by placing Russian 1835 on the opposite side of Estonian and Latvian SLs, while the *n*-dimensional CNets show three-dimensional boxes (when using a cut-off of BS ≥ 15; electronic supplementary material, 4.3, figure S6).

Figure 4 shows stacked NNets including SLs from specific time periods: the lowest NNet in the figure with SLs up to 1840; the middle from 1808 to the late twentieth century; and the uppermost NNet including

**Table 1.** Non-parametric bootstrapping (BS) support for neighbourhood-defining splits of main groups (highlighted in figure 3). ML, maximum likelihood; ASC, corrected for ascertainment bias; UNC, uncorrected for ascertainment bias; NJ, neighbour-joining; P, parsimony.

| SL group | MLBS | | NJBS | PBS |
|---|---|---|---|---|
| | ASC | UNC | | |
| Austrian-origin | <15 | <15 | 24 | 21 |
| British-origin | 99 | 99.5 | 99.8 | 91 |
| French-origin | <15 | <15 | 42 | 16 |
| Afghan-Jordanian | 99.9 | 99.6 | 66 | 98 |
| Russian[a] | 18 | 17 | 40 | 39 |
| Polish[a] | 93 | 90 | 99 | 98 |
| Spanish | 84 | 85 | 71 | 49 |
| Swedish | 99 | 99 | 97 | 98 |

No alternative found with BS ≥ 15.

[a]Not including Latvian SL (see electronic supplementary material, 4.3).

SLs from the mid-twentieth century to present. The bottom graph demonstrates the substantial diversity among MAs by the early nineteenth century, with three main clusters: Austrian-origin, French-origin and Spanish. Based on these graphs, the Austrian-origin group diversified further during the second half of the nineteenth century (middle graph), while the French SL MA was dispersed largely unmodified to the Americas. The Russian group is closest to the Austrian-origin group (the early Austrian SL MAs from 1786 and 1823, as well as their close relatives, the early German 1820 and Hungarian 1827 MAs) and most distant to the French-origin group. The third main distinct cluster in the bottom graph is the Spanish group. The Swedish group, appearing first in the middle graph, is already unique by 1866. In the topmost graph, the overall picture remains the same, with a few exceptions. First, Polish SL, which is found in the Austrian-origin group in the middle graph, is positioned between the Austrian-origin and Russian groups and forms a cluster with Lithuanian SL. Second, contemporary Norwegian SL separates from the middle of the graph and is no longer grouped closely with Danish or Icelandic SLs. Third, Austrian, Danish and German SLs, earlier examples of which were found in the Austrian-origin group, are grouped closely with the International Sign MA and with American SL.

The results of the EDA indicate that each major group goes back to an independent founding event: there is little to no support for the deepest splits; the centre of the overall NNet in figure 3 is web-like; and there are three distinct clusters of the earliest SLs in figure 4. Contemporary Spanish and Catalan SLs are direct derivates of the oldest MAs in our dataset, Yebra 1593 and Bonet 1620, while the French- and Austrian-origin groups constitute the two main independent traditions in continental Europe. Based on the results, we infer that slightly modified versions of the French SL MA were dispersed into the Americas, with the American SL MA later forming the basis for the International Sign MA, which had a homogenizing effect on several European MAs. International Sign possibly affected Norwegian SL and fully replaced the Austrian-origin MA forms in Danish, German and Austrian SLs. Standardization also influenced internal relationships within the French-origin group: we observe a 'taxonomic turnover' with the original SLs in Europe and the New World (closest to French 1800) being replaced by versions very similar (International Sign subgroup; see figure 4) or more similar (contemporary French subgroup) to American SL and International Sign than to the French original, with the exception of contemporary Dutch (unique development), as well as Greek and Irish SLs (still closest to original eighteenth/nineteenth century French).

Contemporary Icelandic SL is a direct derivate from the Danish subgroup within the Austrian-origin group; the same holds for Norwegian SL, which started to strongly deviate from the closely related Icelandic SL in the second half of the twentieth century. The Russian group can be linked historically to the Austrian-origin group (see the bottom-most NNet in figure 4), but likely underwent substantial restructuring in the adaptation from representing a Latin-based alphabet to Cyrillic. The Estonian and Latvian SL MAs are, to a lesser degree, twentieth century derivates, with more links to the contemporary Russian group than to the Russian MA from 1835 (see also electronic supplementary

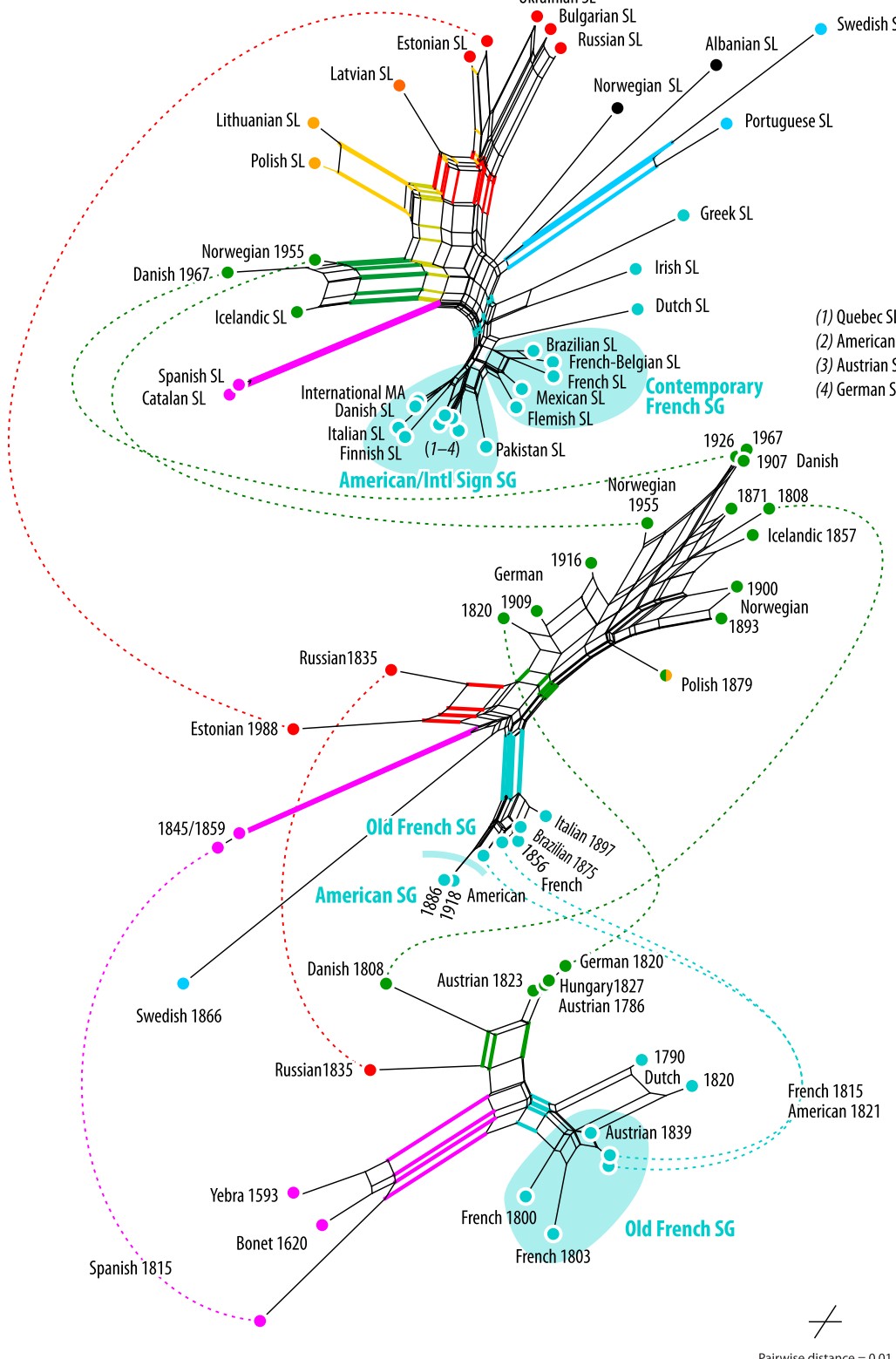

**Figure 4.** Time-/taxon-filtered stacked Neighbour-nets, based on the same distance matrix used for figure 3. Bottom NNet: SLs up to 1840; middle NNet: 1808–late twentieth century; top NNet: mid-twentieth century–present. Abbreviation: SG denotes potential subgroups within the French-origin group. MAs included in two subsequent NNets are connected by dashed lines.

material, 4.3). Swedish constitutes an isolated, mainly unique tradition and is the basis of contemporary Swedish and Portuguese SLs. Although our dataset does not include any historical examples of the British-origin and Afghan-Jordanian groups, we conclude based on the EDA that these groups

constitute isolated traditions that evolved independently (British-origin) or largely independently (Afghan-Jordanian) of the European groups.

## 4. Discussion

Figure 5 shows five European lineages and their hypothesized dispersal from the late sixteenth to the late nineteenth century. Hypothesized dispersal pathways in figure 5 synthesize the results in §3 with dates from historical sources for the establishment of schools for the deaf, known migrations of signers, and the pre-eighteenth century publications of MAs in Spain (see electronic supplementary material, 2.3, table S3). The earliest MAs in our sample, Yebra 1593 and Bonet 1620, are typically identified as the ancestors of most one-handed MAs in the world today [54]. We argue that, while communities in Austria, France and Spain all used these early Spanish sources to form MAs, they did so independently of one another, forming three separate SL lineages. The Yebran and Bonetian MAs, together with other Spanish MAs in the sample (Spanish 1815, 1845, 1859, contemporary Spanish and Catalan SLs), constitute a Spanish lineage. The NNet in figure 3 gathers these MAs in a highly coherent group with moderate to high BS support (table 1) without alternatives (see CNets provided in electronic supplementary material, 4.3 and original files included). Because there existed no large-scale deaf educational institutions in Spain until the early nineteenth century, it seems unlikely that the two early MAs were used widely prior to that period, particularly in signing communities. There are 22 letters represented by forms in the Yebran and Bonetian MAs. Fifteen forms in the Spanish MA from 1815 are unmodified from the Yebran and Bonetian MAs, while other forms show small modifications from the early sources (see electronic supplementary material, 4.4, figure S7 for a comprehensive overview of differences for the earliest MAs in Spain, France and Austria compared with the Bonetian and Yebran MAs). The 1815 form representing ⟨d⟩ differs slightly in index finger extension compared with Bonet; 1815 forms for ⟨m⟩ and ⟨n⟩ differ only in the separation of the extended fingers compared with Bonet and in thumb position compared with Yebra. That the early MAs were not used and transmitted within a signing community until the early nineteenth century, and were thus unaffected by evolutionary processes connected with their usage and transmission, may explain the stability of many MA forms over almost 200 years from Bonet's MA in 1620 to the Spanish MA in 1815.

Multiple sources report that the founder of deaf education in France, de l'Épée in Paris, had become aware of Bonet's (1620) *Reduction de las Letras* and its MA by the late eighteenth century [8,69], suggesting that he may have used Bonet's MA as a source in forming the French MA. We argue here that the French MA drew on Yebra's and Bonet's MAs, but did not evolve directly from a shared Spanish–French origin. First, we conclude that the French MA drew on the early Spanish sources directly, and not the later sources from 1815 and following. Where the original and later Spanish forms differed, the French MA either kept the original form unmodified (e.g. in the handshapes representing ⟨m⟩, ⟨n⟩ and ⟨u⟩) or modified the original and not the later Spanish form (e.g. in the handshape representing ⟨q⟩). Second, independent formation of the French MA is suggested by the clear separation of the French and Spanish groups in figure 3 and of the earliest French and Spanish MAs in figure 4. The Paris Institute had been in existence for decades before the school in Madrid was founded, making it unlikely that there could have been a common Spanish–French basis from which the French MA evolved. Moreover, while the Spanish MA from 1815 shows little innovation compared to Yebra and Bonet, many differences are observable between the Yebran and Bonetian MAs and the earliest French MA in our sample (French 1800), in which just 11 of 22 forms remained unmodified and forms were added representing new letters ⟨j⟩, ⟨k⟩, ⟨v⟩ and ⟨w⟩. Thus, in adapting the early Spanish sources, the signing community in Paris changed them substantially, both consciously by innovating new forms and through usage in a community.

The other main continental European lineage is the Austrian-origin lineage. Some scholars have claimed that Austrian and French SLs share a common origin [19]. A large number of uniquely shared innovations (*synapomorphies* in biology) would support such a conclusion, indicating a unified basis from which both MAs later diverged. Shared innovations would be, in this case, those forms that differed from Spanish sources but that were uniquely shared in French and Austrian MAs. In fact, we find relatively few such potentially derived forms between the earliest Austrian and French MAs, including those representing ⟨d⟩, ⟨e⟩, ⟨l⟩, ⟨r⟩ and ⟨w⟩. Forms for ⟨f⟩ and ⟨v⟩ were shared by Austrian 1786 and French 1800, but also by early unrelated SLs, such as Swedish 1866.

That these possible synapomorphies across Austrian and French MAs are too few is reflected in the lack of a French–Austrian neighbourhood *trunk* in the NNet of the earliest MAs in the sample (figure 4). If Austrian had indeed evolved from a French basis, the network should show a prominent fan-like

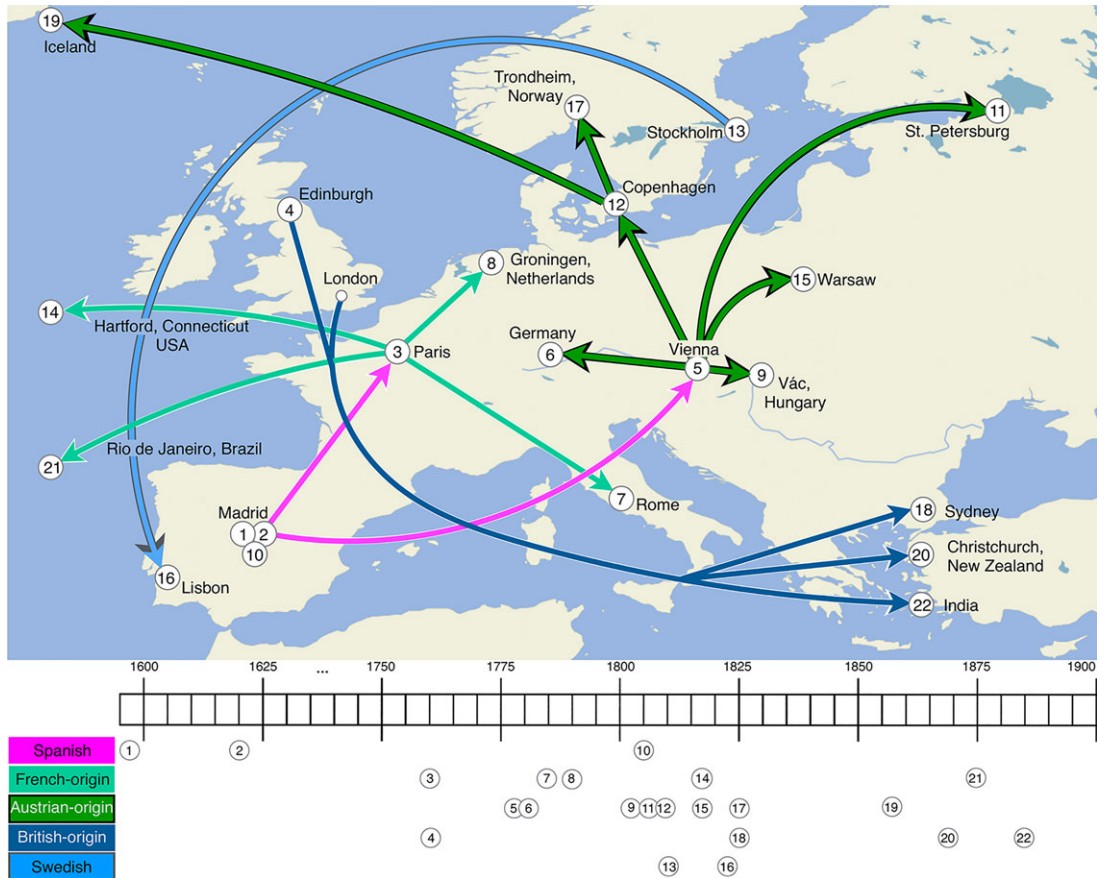

**Figure 5.** Hypothesized dispersal of European SLs from late sixteenth to late nineteenth century, based on results in §3. Colour-coding reflects five hypothesized lineages: Spanish, French-origin, Austrian-origin, British-origin, Swedish. Timeline reflects approximate years of first transmission of SLs, coinciding with establishment of schools for the deaf or migrations of signers (except in the cases of 1 and 2, which track publication of earliest MAs in sample).

structure including both MAs, with the oldest MAs in the middle and the most derived within each group as wings towards either side of the fan. The potential shared innovations may be better characterized as early borrowings due to minimal contact between French and Austrian signing communities; or as convergent evolution, in which handshape forms independently evolved in similar ways either to iconically match the forms of similar graphemes, or because forms were selected for their conferred articulatory and perceptual advantages.

Finally, compare the positions of the Austrian, French and Spanish groups in figure 3 to the positions of other languages thought to have evolved from French SL. For example, we know that deaf French educators helped establish deaf education in the USA [15], Brazil [16] and Mexico [17]; and those MAs remain topologically close to contemporary French SL. In figure 4, the earliest American MA in the sample (American 1821) has diverged minimally from the chronologically closest French MA from 1815. By contrast, early Austrian, French and Spanish MAs are found in differing neighbourhoods.

The results support classifying Danish as a sublineage of the Austrian-origin lineage. Forms representing ⟨c⟩, ⟨g⟩, ⟨h⟩, ⟨o⟩, ⟨p⟩ and ⟨q⟩ in the earliest Danish MA from 1808 are shared with Austrian but not French. By contrast, none of the forms in the earliest Danish MA from 1808 are unambiguously French in origin. These patterns are reflected in the bottom graph in the NNet in figure 4; Danish 1808 is topologically closest to the early Austrian-origin group languages. The early Danish MA shows several innovated handshapes, such as those representing ⟨d⟩, ⟨f⟩, ⟨k⟩, ⟨s⟩, ⟨u⟩, ⟨v⟩, ⟨w⟩, ⟨y⟩ and ⟨z⟩, as well as new forms for the Danish letters ⟨æ⟩ and ⟨ø⟩. Thus, there is support for classifying Danish as a separate lineage based on similar argumentation to that used above. In contrast to the Austrian, French and Spanish cases above, however, early Danish and early Austrian MAs are consistently found in close topological proximity in the network analyses under various methods and optimality criteria.

While the results support classifying Russian as a separate lineage created using Austrian sources, any interpretation of the results with respect to Russian SL is complicated by substantial adaptation and restructuring of an existing MA to represent the Cyrillic alphabet. Similarities between Russian SL and

early Austrian MAs are reflected in the NNet of early MAs in figure 4, which places Russian 1835 closest to the Austrian-origin group languages compared to other groups. Because of mismatches between the two alphabets, however, the Austrian MA underwent substantial restructuring to represent graphemes in Cyrillic. New forms were invented for some Cyrillic letters not found in the Latin alphabet (e.g. ⟨б⟩); other forms were dropped (e.g. the Austrian form representing ⟨d⟩); and some forms were used to represent Cyrillic letters that appear similar to Latin letters, but which occupy different positions in the alphabet (e.g. Cyrillic ⟨Я⟩). One consequence of this mismatch between alphabets is that there are fewer cross-alphabet comparanda between Latin- and Cyrillic-representing MAs. In addition, as discussed in §2, there are serious methodological challenges in deciding which forms should be compared, with the resulting possibility that some connections between sources and adaptations cannot be recovered. Thus, differences between Latin and Cyrillic alphabets may cause phylogenetic methods to overestimate the distance between the Austrian-origin and Russian groups. Conversely, because of iconicity, our approach in this comparison may, in general, underestimate distances between MAs representing the same type of alphabet. Many MA forms represent letters iconically, for example, by matching the shape of the letter with the shape of the hand. MAs representing typologically similar alphabets are more likely to converge on handshapes, evolving independently to similar forms. It is also possible that iconic MA forms evolve more slowly or resist changes in order to maintain the iconic link between the handshape and letter represented.

The British-origin group forms an independent lineage. The NNet in figure 3 shows a clear split of this group from the centre of the graph. While the links between British SL and SLs in Commonwealth countries are relatively well documented [23], less is known about how the British-origin MA came to other SLs in the group, namely, Czech, Croatian and Turkish SLs. Zeshan [70] reports the influence of British pedagogical methods in Turkey in the early 1950s. Both Kuhn *et al.* [71] for Croatian SL and Hudáková [72] for Czech SL report that one- and two-handed MAs are in use in Croatia and the Czech Republic: in Croatia, the two-handed alphabet is thought to be older, while the opposite may be true in the Czech Republic. Future research could uncover historical examples of MAs from these SLs that can help to clarify their connections to the British-origin group. Lexical investigations may also shed more light.

Finally, the Swedish group, which consists of only historical and contemporary Swedish SL, as well as Portuguese SL, form a separate lineage. The edge bundle separating the Swedish group from the centre of the graph in figure 3 has high BS support (table 1). The earliest example of the Swedish MA in our sample (Swedish 1866) is unique, with a long edge separating it from the centre of the middle graph in figure 4. Bergman & Engberg-Pedersen [20] suggest that Per Aron Borg, the founder of the first school for the deaf in Stockholm in 1809, though aware of de l'Épée's work in the Paris Institute, may not have been familiar with the MA in use in France when he created the Swedish MA. However, there are similarities between the Swedish MA from 1866 and MAs used in France and other parts of Europe, such as in handshapes representing ⟨c⟩, ⟨f⟩, ⟨k⟩, ⟨l⟩, ⟨m⟩, ⟨n⟩, ⟨o⟩, ⟨u⟩ and ⟨v⟩. Thus, while the results support the conclusion that the new Swedish MA was created mainly in isolation from other lineages, Borg may have borrowed a limited number of handshapes known widely in Europe. That he may not have had a preference for handshapes from any existing MA is indicated by the general lack of branch support for an alternative that would place the Swedish lineage as sister to any other group. The only somewhat similar MA, according to the results, is Albanian SL (electronic supplementary material, SI 4.2, figure S6), which also lacks any clear affinity to any of the major groups.

# 5. Conclusion

Despite their relevance to our understanding of human linguistic diversity and to theories of language change, the evolutionary histories of the world's SLs have not, until now, been studied using state-of-the-art methods. We have shown that computational phylogenetic methods can be applied to MA data to uncover new insights into the evolutionary histories of SLs, to generate new hypotheses about their relationships, and to better understand the evolutionary processes that have shaped the diversity of contemporary SLs. Our analysis supports some aspects of existing SL classifications, such as relationships among French SL and Dutch, Flemish and French-Belgian SLs, as well as SLs of the Americas, including American, Brazilian, Mexican and Quebec SLs. In addition, our results support previous analyses of the relationships of BANZSL languages, and use language data to confirm known historical institutional connections among Swedish and Portuguese SLs, as well as Danish, Icelandic and Norwegian SLs. At the same time, our results add complexity to the overall picture of SL relationships, in particular to our understanding of the evolution of Austrian, French and Spanish SLs from early sources. Our discussion of the independent establishment of SL lineages points to a characterization of

similarities across lineages as primarily horizontal, and not due to descent from a common ancestor, while within-lineage diversification does appear to be characteristic vertical in many cases. Finally, our comparison has generated new hypotheses about relationships among Austrian, Danish, Polish and Russian SLs, as well as other SLs in those groups. We anticipate that future studies of lexical data may contradict our phylogeny based on MAs, in particular for SLs that adopted the International Sign MA, because this adoption did not likely affect a language's lexicon to any great extent. Notwithstanding these limitations, we suggest that our analysis be taken in future research as the best available phylogenetic classification of these SLs.

Data accessibility. The complete database of annotated MAs has been curated on GitHub (https://github.com/lexibank/powerma) and archived with Zenodo (https://zenodo.org/record/3564465, Version v1.0.2). The code that we used to analyse the data along with the results of the analysis have also been curated on GitHub (https://github.com/lingpy/sign-language-evolution-paper) and archived with Zenodo (https://zenodo.org/record/3564484, Version v1.0.1).
Authors' contributions. J.M.P. and J.M.L. initiated the study, designed the database and graphics 1, 2 and 5, and developed methods for data coding. J.M.L. programmed the database system. J.M.P. collected and coded the data. G.W.G. carried out the phylogenetic analysis and annotated the network graphs and consensus networks. J.M.P. and G.W.G. interpreted the results and wrote the first draft. All authors read the last draft and agree on its contents.
Competing interests. We declare we have no competing interests.
Funding. J.M.L. has received funding from the European Research Council (ERC) under the European Union's Horizon 2020 research and innovation programme (grant agreement no. 715618, *Computer-Assisted Language Comparison*, https://digling.org/calc).
Acknowledgements. We thank Tiago Tresoldi for sharing initial ideas on this project, Harald Hammarström for providing information on proposed sign language classifications, and Russell D. Gray and Richard P. Meier for comments on an earlier version of this paper.

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
