## [Reviewer comments · Royal Society Open Science]

Review History

RSOS-191100.R0 (Original submission)

Review form: Reviewer 1

Is the manuscript scientifically sound in its present form?

Yes

Are the interpretations and conclusions justified by the results?

Yes

Is the language acceptable?

Yes

Do you have any ethical concerns with this paper?

No

Have you any concerns about statistical analyses in this paper?

No

Recommendation?

Major revision is needed (please make suggestions in comments)

Comments to the Author(s)

The paper presents a novel study in which quantitative methods are used to resolve the relationships and dispersal of sign languages. The work is of scientific interest and in general well conducted. The paper would, however, benefit of a notable restructuring of text in the way that the earlier knowledge and hypotheses of the dispersal of sign languages would be presented thoroughly already in the introduction and not mainly in the discussion as it is now. This way it would be clearer what is known beforehand based on the earlier literature and what is found in this study, as in the current version this distinction is not always clear. Further and more specific comments are listed below.

1. Introduction

p2 r 8-9: I find the first sentence of the introduction "Linguistic analyses of the world's sign languages --" a bit confusing as now it sounds that it is the linguistic analyses that have shown that there can be a sign language that can be used as a mean of communication. However, as SLs have been used for communication for centuries, I would see that as a more important sign of the fact that SLs can be used in communication. Therefore, some changes should be made to the beginning of the first paragraph.

p2 r 11: It would be good to have some notions about the earlier/earliest existence of SL before going to the 18th and 19th century's conventional and widespread SLs, especially as these older ones are also part of your dataset (Yebra and Bonet, are there others?).

p2 r 20: Why there is a question mark after the reference 5?

p2 r 21-24: The spread of French SL to Netherlands, US and Brazil is mentioned in the text. It would be good if this kind of 'from - to' information would also be mentioned clearly about the other European SLs. It would serve as a proper introduction to the known spread and possible groups of SLs.

As the spread of French SL mentioned here matches with the lines spreading from France in Figure 5, it would be easier follow if this would be the case also with the other European SLs. Actually, if there is information about the possible routes of spread of SLs, these could be used as hypotheses for the analyses. This kind of re-structuring of the introduction would give it a more straightforward goal than the current "investigating evolutionary histories of SLs".

In general: There is a mismatch with the reference numbers in the text and those in the reference list (e.g. I assume the references about recent work with phylogenetic methods should be 22 and 23 instead of 21 and 22 as it is now). It needs to be checked where this mismatch starts and the mismatch should be corrected.

There should be more general information in the introduction about sign languages and e.g. what are the 'manual alphabets' mentioned in row 43. There is more explanation about these in the materials and methods section (starting p3 r37), but I think the basics should be already in the introduction.

Furthermore, there should be information about how many different SLs there is in the world so does the 76 manual alphabets cover all the SLs of the world or are they a sample of them? If those are a sample, how and why exactly those where chosen for the study?

2. Materials and Methods

p2 r 57: Please explain in more detail what kind of a transcription HamNoSys does. Transcribes pictures of hand to different (sets of) symbols? Or something else?

p2 r 58: Please explain in more detail how EDICTOR evaluates morphological similarity.

p3 r58-p4 r10: It should be clarified what is the specific issue that has been the topic of debate concerning the linguistic status of MAs and their relationship to signed and written language. Do some SLs use MAs more than some others?

Concerning fingerspelling, I don't see what is the problem (or risk for losing the historical signal) if children learn the MAs before they are able to read. Or is the point that the MAs they learn as a kid are not the official ones? Now when it states "However, usage of fingerspelling varies cross-linguistically and across signers within the same signing community", it does not yet tell whether they vary in frequency in usage or in the MAs they use. The paragraph should be clarified to answer these questions.

2.2 Character coding

p4 r18: "We considered only handshapes and movements for determining similarity of MA forms." In page 3 you say that an individual form of MA is comprised e.g. of a handshape with a particular spatial orientation. Is this spatial orientation taken into account when determining the similarity of MA forms or only the handshapes? This should be clarified in the text.

p4 r28: You write "In the example just mentioned, we compared handshapes representing Cyrillic B with Latin B and not V--" However, in the example above it is not said that Cyrillic B is compared with Latin B but instead only mentioned that decision needs to be made whether the comparison is done with Latin B or V. Thus, please rephrase.

p4 r31. "When graphemic forms across two alphabet types are similar as in the case of Latin and Cyrillic just mentioned, the form of the grapheme typically becomes the basis of comparison." In this the part "typically becomes" is quite vague. It should be specified in which cases the form of the grapheme was not used even those were similar.

p4 r36-37: Explanation about character mapping and what can be achieved with it should be elaborated.

p4 r41. Is 'concept' the basic unit in your data (cf. 'meaning' in basic vocabulary lists)? If yes, the explanation about the example would be clearer if it would start by explaining that it is about "concept g" and only after that the divisions within it.

p4 r50-57: Was the similarity obtained automatically by EDICTOR tool or was it based on subjective human evaluation? Was it always clear which handshapes are assigned similar and which not? These issues should be elaborated in the text.

2.3. Phylogenetic analysis

Please elaborate in this section which different sets of analyses with which data were done (whole data set, divisions into time layers etc.). What is the purpose of using both NNets and C Nets? This should be mentioned also in the main text and not only in the Supplementary Information. Please also refer to Supplementary Information for more information and explain which results are shown in the Results section of the main text and which in the SI.

3. Results

p6 r29: It would be clearer if the third group (British) would be presented only after the Austrian and French groups are clarified and when starting to explain about the Afghan-Jordanian group. Figure 3. It would be good to have the bootstrap values of different groups also in the plot as some of the main groups are clearly supported while others are not.

Also, in Figure 3, some SLs are marked with a colored circle that differs from the time scale given in the figure: Russian 1835 is red, French 1800 is blue, Yebra 1593 is purple and Austrian 1786 is green. The reason for the differing coloration of these SLs should be mentioned in the Figure legend or changed so that the coloring follows the given time slots.

p8 r31: It should be mentioned in the materials and methods that NNets are done for different time periods separately. It should also be said why some historical samples were excluded from the analyses as now these are just mentioned in the end of the results section.

p8 r32: The time slots are different in the text and in the legend of Figure 4. They should be in line with each other.

p8 r37: Please add with a reference when did the French SL MA disperse to the Americas?

p8 r53-59: Explanation about International Sign and its homogenizing effect on European SLs (with references) should be moved to the introduction.

Figure 4. It would be good to have bootstrap values also in Figure 4, at least for the edge bundles of the main groups. In addition, in the bottom NNet the split connecting French group with German 1820, Hungary 1827 and Austrian 1786 looks longer than then one splitting only the French group. Would there be something interesting to discuss about? It should also be noted what the dashed lines denote.

Figure 5. The two greens and the two shades of blue look very similar in the Figure. It would be good to give a bit more different colours to these.

In the legend it says "Timeline reflects approximate years of first transmission of SLs, typically coinciding with establishment of schools." To what is the first transmission time of SL based on or is it the time of the establishment of the school?

4. Discussion

p10 r57-58: The first sentence in Discussion is an example of a sentence of which I am not completely sure what it is based on. If it is based on the time layer analysis and Figure 4, it is problematic as those analyses did not include British and Swedish group, which nevertheless are part of Figure 5.

p11 r19: "While the French MA was clearly formed using Spanish sources--." It is again unclear whether this is based on earlier literature or the results of this study. Please make this division clear here and also in other parts of the text.

p11 r30: "--in which just eleven forms remained unmodified--" Eleven forms out of how many?

p11 r31: "In adapting the early Spanish sources, the signing community in Paris changed them substantially ... , which resulted only minimal modifications to the existing forms." It is now a bit unclear whether there were substantial or minimal modifications to the early Spanish sources. This should be clarified.

p11 r33-36: Could the sentence starting "It is clear that the--" be said in a more straightforward way as the current version is quite complex.

p11 r37-41: Example of a sentence that should be already in the introduction and not only in the discussion. The following sentence "Perhaps because --" would also make an appropriate hypothesis what to study and should also be mentioned already in the introduction.

p12 r45-55: It should be mentioned already in the introduction that there is also a 2 handed MA and that for this reason the British group may be expected to appear as a group of its own.

p12 r46: Reference is denoted only by a question mark.

In general: It would be good to discuss shortly the possible role of iconicity to the results as perhaps some MAs are more iconic and thus less likely to change.

Conclusion

It would be good to summarise in conclusions what were the main findings and how they differed from what was expected.

p13 r31-32: Is there a word missing in "--we suggest that our analysis [could/should] be taken in future research --"?

Supplementary Information

1. Organisation of the supplementary material

The first link given in the text does not work.

2.2 Historical manual alphabets

The word 'in' is twice one after another in the sentence.

3.2 Coding

Apparently, the year of the historical Estonian SL is missing both from the table and from the text, or I do not see how the concept of h could be both the narrow concept of h and n in contemporary Estonian.

4.2 Character mapping

Please elaborate how the character mapping was done and rephrase the section and subsections to make it more like the main text, e.g. rephrase the text in the way that it starts with how the median networks were reconstructed and not with reference to figures. Also, add legends to the figures in sections 4.2.1, 4.2.2 and 4.2.3 and give these figures appropriate figure numbers which you refer in the SI text.

More specifically, in page 7, row 4 the replacement of Dutch version by Austrian is denoted as ("At->Du") whereas in the figure it is other way around. This should be changed so that it is correct in both.

4.2.1. It is not immediately clear to which the "list" refers to. This should be made clearer.

Decision letter (RSOS-191100.R0)

09-Aug-2019

Dear Dr List,

The editors assigned to your paper ("Evolutionary dynamics in the dispersal of sign languages") have now received comments from reviewers. We would like you to revise your paper in accordance with the referee and Associate Editor suggestions which can be found below (not including confidential reports to the Editor). Please note this decision does not guarantee eventual acceptance.

Please submit a copy of your revised paper before 01-Sep-2019. Please note that the revision deadline will expire at 00.00am on this date. If we do not hear from you within this time then it will be assumed that the paper has been withdrawn. In exceptional circumstances, extensions may be possible if agreed with the Editorial Office in advance. We do not allow multiple rounds of revision so we urge you to make every effort to fully address all of the comments at this stage. If deemed necessary by the Editors, your manuscript will be sent back to one or more of the original reviewers for assessment. If the original reviewers are not available, we may invite new reviewers.

- Data accessibility

<http://datadryad.org/submit?journalID=RSOS&manu=RSOS-191100>

- **Competing interests**

- **Authors' contributions**

- **Acknowledgements**

- **Funding statement**

on behalf of Professor Kevin Padian (Subject Editor)
openscience@royalsociety.org

Associate Editor's comments to the Author:

Unusually, and owing to the difficulty had in finding sufficient suitable reviewers, the Editors are opting to make their decision on the basis of one reviewer's report. The report offers a range of recommendations for improvement, and the referee will be invited to assess your revision, so we urge you to take due care in preparing the revision to ensure you fully address their concerns - please ensure you also include a full point-by-point response with your revision, and be aware that if you do not fully satisfy the reviewer that the paper is ready for publication, we may not be able to consider the paper further for publication. Good luck.

Reviewers' Comments to Author:

Reviewer: 1

Comments to the Author(s)

The paper presents a novel study in which quantitative methods are used to resolve the relationships and dispersal of sign languages. The work is of scientific interest and in general well conducted. The paper would, however, benefit of a notable restructuring of text in the way that the earlier knowledge and hypotheses of the dispersal of sign languages would be presented thoroughly already in the introduction and not mainly in the discussion as it is now. This way it would be clearer what is known beforehand based on the earlier literature and what is found in this study, as in the current version this distinction is not always clear. Further and more specific comments are listed below.

1. Introduction

p2 r 8-9: I find the first sentence of the introduction "Linguistic analyses of the world's sign languages --" a bit confusing as now it sounds that it is the linguistic analyses that have shown that there can be a sign language that can be used as a mean of communication. However, as SLs have been used for communication for centuries, I would see that as a more important sign of the fact that SLs can be used in communication. Therefore, some changes should be made to the beginning of the first paragraph.

p2 r 11: It would be good to have some notions about the earlier/ earliest existence of SL before going to the 18th and 19th century's conventional and widespread SLs, especially as these older ones are also part of your dataset (Yebra and Bonet, are there others?).

p2 r 20: Why there is a question mark after the reference 5?

p2 r 21-24: The spread of French SL to Netherlands, US and Brazil is mentioned in the text. It would be good if this kind of 'from - to' information would also be mentioned clearly about the other European SLs. It would serve as a proper introduction to the known spread and possible groups of SLs.

As the spread of French SL mentioned here matches with the lines spreading from France in Figure 5, it would be easier follow if this would be the case also with the other European SLs. Actually, if there is information about the possible routes of spread of SLs, these could be used as hypotheses for the analyses. This kind of re-structuring of the introduction would give it a more straightforward goal than the current "investigating evolutionary histories of SLs".

In general: There is a mismatch with the reference numbers in the text and those in the reference list (e.g. I assume the references about recent work with phylogenetic methods should be 22 and 23 instead of 21 and 22 as it is now). It needs to be checked where this mismatch starts and the mismatch should be corrected.

There should be more general information in the introduction about sign languages and e.g. what are the 'manual alphabets' mentioned in row 43. There is more explanation about these in the materials and methods section (starting p3 r37), but I think the basics should be already in the introduction.

Furthermore, there should be information about how many different SLs there is in the world so does the 76 manual alphabets cover all the SLs of the world or are they a sample of them? If those are a sample, how and why exactly those where chosen for the study?

2. Materials and Methods

p2 r 57: Please explain in more detail what kind of a transcription HamNoSys does. Transcribes pictures of hand to different (sets of) symbols? Or something else?

p2 r 58: Please explain in more detail how EDICTOR evaluates morphological similarity.

p3 r58-p4 r10: It should be clarified what is the specific issue that has been the topic of debate concerning the linguistic status of MAs and their relationship to signed and written language. Do some SLs use MAs more than some others?

Concerning fingerspelling, I don't see what is the problem (or risk for losing the historical signal) if children learn the MAs before they are able to read. Or is the point that the MAs they learn as a kid are not the official ones? Now when it states "However, usage of fingerspelling varies cross-linguistically and across signers within the same signing community", it does not yet tell whether they vary in frequency in usage or in the MAs they use. The paragraph should be clarified to answer these questions.

2.2 Character coding

p4 r18: "We considered only handshapes and movements for determining similarity of MA forms." In page 3 you say that an individual form of MA is comprised e.g. of a handshape with a particular spatial orientation. Is this spatial orientation taken into account when determining the similarity of MA forms or only the handshapes? This should be clarified in the text.

p4 r28: You write "In the example just mentioned, we compared handshapes representing Cyrillic B with Latin B and not V--" However, in the example above it is not said that Cyrillic B is compared with Latin B but instead only mentioned that decision needs to be made whether the comparison is done with Latin B or V. Thus, please rephrase.

p4 r31. "When graphemic forms across two alphabet types are similar as in the case of Latin and Cyrillic just mentioned, the form of the grapheme typically becomes the basis of comparison." In this the part "typically becomes" is quite vague. It should be specified in which cases the form of the grapheme was not used even those were similar.

p4 r36-37: Explanation about character mapping and what can be achieved with it should be elaborated.

p4 r41. Is 'concept' the basic unit in your data (cf. 'meaning' in basic vocabulary lists)? If yes, the explanation about the example would be clearer if it would start by explaining that it is about "concept g" and only after that the divisions within it.

p4 r50-57: Was the similarity obtained automatically by EDICTOR tool or was it based on subjective human evaluation? Was it always clear which handshapes are assigned similar and which not? These issues should be elaborated in the text.

2.3. Phylogenetic analysis

Please elaborate in this section which different sets of analyses with which data were done (whole data set, divisions into time layers etc.). What is the purpose of using both NNets and CNets? This should be mentioned also in the main text and not only in the Supplementary Information. Please also refer to Supplementary Information for more information and explain which results are shown in the Results section of the main text and which in the SI.

3. Results

p6 r29: It would be clearer if the third group (British) would be presented only after the Austrian and French groups are clarified and when starting to explain about the Afghan-Jordanian group. Figure 3. It would be good to have the bootstrap values of different groups also in the plot as some of the main groups are clearly supported while others are not.

Also, in Figure 3, some SLs are marked with a colored circle that differs from the time scale given in the figure: Russian 1835 is red, French 1800 is blue, Yebra 1593 is purple and Austrian 1786 is green. The reason for the differing coloration of these SLs should be mentioned in the Figure legend or changed so that the coloring follows the given time slots.

p8 r31: It should be mentioned in the materials and methods that NNets are done for different time periods separately. It should also be said why some historical samples were excluded from the analyses as now these are just mentioned in the end of the results section.

p8 r32: The time slots are different in the text and in the legend of Figure 4. They should be in line with each other.

p8 r37: Please add with a reference when did the French SL MA disperse to the Americas?

p8 r53-59: Explanation about International Sign and its homogenizing effect on European SLs (with references) should be moved to the introduction.

Figure 4. It would be good to have bootstrap values also in Figure 4, at least for the edge bundles of the main groups. In addition, in the bottom NNet the split connecting French group with German 1820, Hungary 1827 and Austrian 1786 looks longer than then one splitting only the

French group. Would there be something interesting to discuss about? It should also be noted what the dashed lines denote.

Figure 5. The two greens and the two shades of blue look very similar in the Figure. It would be good to give a bit more different colours to these.

In the legend it says "Timeline reflects approximate years of first transmission of SLs, typically coinciding with establishment of schools." To what is the first transmission time of SL based on or is it the time of the establishment of the school?

4. Discussion

p10 r57-58: The first sentence in Discussion is an example of a sentence of which I am not completely sure what it is based on. If it is based on the time layer analysis and Figure 4, it is problematic as those analyses did not include British and Swedish group, which nevertheless are part of Figure 5.

p11 r19: "While the French MA was clearly formed using Spanish sources--." It is again unclear whether this is based on earlier literature or the results of this study. Please make this division clear here and also in other parts of the text.

p11 r30: "--in which just eleven forms remained unmodified--" Eleven forms out of how many?

p11 r31: "In adapting the early Spanish sources, the signing community in Paris changed them substantially ... , which resulted only minimal modifications to the existing forms." It is now a bit unclear whether there were substantial or minimal modifications to the early Spanish sources.

This should be clarified.

p11 r33-36: Could the sentence starting "It is clear that the--" be said in a more straightforward way as the current version is quite complex.

p11 r37-41: Example of a sentence that should be already in the introduction and not only in the discussion. The following sentence "Perhaps because --" would also make an appropriate hypothesis what to study and should also be mentioned already in the introduction.

p12 r45-55: It should be mentioned already in the introduction that there is also a 2 handed MA and that for this reason the British group may be expected to appear as a group of its own.

p12 r46: Reference is denoted only by a question mark.

In general: It would be good to discuss shortly the possible role of iconicity to the results as perhaps some MAs are more iconic and thus less likely to change.

Conclusion

It would be good to summarise in conclusions what were the main findings and how they differed from what was expected.

p13 r31-32: Is there a word missing in "--we suggest that our analysis [could/should] be taken in future research --"?

Supplementary Information

1. Organisation of the supplementary material

The first link given in the text does not work.

2.2 Historical manual alphabets

The word 'in' is twice one after another in the sentence.

3.2 Coding

Apparently, the year of the historical Estonian SL is missing both from the table and from the text, or I do not see how the concept of h could be both the narrow concept of h and n in contemporary Estonian.

4.2 Character mapping

Please elaborate how the character mapping was done and rephrase the section and subsections to make it more like the main text, e.g. rephrase the text in the way that it starts with how the median networks were reconstructed and not with reference to figures. Also, add legends to the figures in sections 4.2.1, 4.2.2 and 4.2.3 and give these figures appropriate figure numbers which you refer in the SI text.

More specifically, in page 7, row 4 the replacement of Dutch version by Austrian is denoted as (“At->Du”) whereas in the figure it is other way around. This should be changed so that it is correct in both.

4.2.1. It is not immediately clear to which the “list” refers to. This should be made clearer.

Author's Response to Decision Letter for (RSOS-191100.R0)

See Appendix A.

RSOS-191100.R1 (Revision)

Review form: Reviewer 1

Is the manuscript scientifically sound in its present form?

Yes

Are the interpretations and conclusions justified by the results?

Yes

Is the language acceptable?

Yes

Do you have any ethical concerns with this paper?

No

Have you any concerns about statistical analyses in this paper?

No

Recommendation?

Accept with minor revision (please list in comments)

Comments to the Author(s)

The manuscript has been improved notably and it is a lot clearer now. I would, however, still have some minor comments which could help to improve the manuscript.

p4 r53-p5 r9. This paragraph is still quite unclear and would benefit from restructuring. First, it would help to say in the end of the first sentence that what is the point of the paragraph. Now the reason (that historically relevant information can be obtained from MA's) is said only in the end of the paragraph. Second, now when the first sentence mentions differences between historical comparative study of vocabulary and study focusing on MA's, the reader assumes that what follows is comparison of these two data types. In the current version, however, only peculiarities of MA system are listed without the comparison with basic vocabulary data. Also, it remained a bit unclear what does it mean in practice that SLs vary with regard to how integral the MA is to the language as a whole so it would be good to give an example of what kind of MAs are more integrated within a SL's lexicon and which are less integrated.

p7 r14. I couldn't find a reference to the SI section 4.1. This should be added to the main text.

p7 r38. Parentheses are missing from the end of (see Figures 5 and 6A-D in SI 4.3).

p8 r51 It is unclear to what the value $BS > 48$ is based on. I couldn't find the value 48 from Table 1.

p10 r12-13. Explanation of how different optimality criteria differ from each other could be removed to methods section.

p10 r17 For clarity, it would be good to refer to the specific SI Figure in which the 3 dimensional boxes concerning the Russian group are shown.

p10 r34-36. As the BS-values were quite low for many of the major groups, including Austrian-origin and French-origin (at least in Figure 3), I find it quite bold to say that each major group goes back to an independent founding event. It should be elaborated how the independent founding events are inferred here. Furthermore, the latter part of the sentence could be clearer if verbs would be added there.

p12 r52-53. "-- we argue that communities in Austria, France and Spain independently formed MAs using the early sources." Does this mean Austria, France and Spain SL communities formed MAs independently from each other or independently from early sources. With current formatting there is a chance of both kind of interpretations and it should be clarified what is meant here.

p12 r53-54. "We suggest that this independent formation supports identifying three separate SL lineages." In this sentence the chain of reasoning seems to be upside down as one would assume that the independent formation would be inferred from the three separate SL lineages and not vice versa. Furthermore, I could not find bootstrap values to the time-/taxon filtered NNets even though a lot of interpretation is based on the groups seen there (especially in the oldest time layer).

p13 15-17. The sentence "Preserved examples in books--" is unclear and should be rephrased.

p13 r21. "-- did not evolve directly from a shared Spanish-French origin." If the point here is to discuss whether French MA evolved from Spanish origin, shouldn't it be just "evolve directly from Spanish origin." It is thus unclear what is meant here with the Spanish-French origin.

Supplementary Information

p7 r1-2. "Handshapes for Latin <n> in contemporary American SL, Austrian SL and German SL were coded as similar and assigned ID 137." When checking what kind of handshapes these three languages have in the SI Figure 1, the version Austrian SL has looks notably different from American and German SLs. Thus it would be good to know whether this is correct and it is only matter of how to read those symbols.

Review form: Reviewer 2

Is the manuscript scientifically sound in its present form?

Yes

Are the interpretations and conclusions justified by the results?

No

Is the language acceptable?

Yes

Do you have any ethical concerns with this paper?

No

Have you any concerns about statistical analyses in this paper?

No

Recommendation?

Accept with minor revision (please list in comments)

Comments to the Author(s)

This paper describes a model of Sign Language dispersion and relatedness on the basis of phylogenetic network methods. This approach is novel and has not been applied to the study of sign languages before, therefore I think there is value in sharing this work, as a first attempt to apply these methods, since others can build on and expand the approach described here. However I also think the paper in its current form has some issues that should be fixed before it is ready for publication. Some of the points I list below have to do with consistency issues that likely emerged between this version and the previous version of the paper. As I understand, a lot more prior knowledge on sign language dispersion from other sources such as historical data has been added to the background and as the first reviewer suggested, the focus of the modeling should be on filling some gaps in the (already quite detailed) picture we get from such other data, instead of claiming that we know nothing and that comparing manual alphabets is going to change this. Some parts of the paper still resonate more with the second goal though.

For instance:

"scholars have been largely unsuccessful in grouping sign languages into monophyletic language families"

As some of the background discussed in the paper also suggests, we actually know quite a lot about how sign languages are related to each other through historical records and cultural information that is preserved since the origins of many currently established sign languages does not go as far back as the spoken languages we speak today.

And another contradiction: "While the exact origins of the MAs used in European educational institutions are unclear" and a bit further we read: "because the creation and transmission of many MAs are well-documented, the data provide a relatively well- understood test case for studying evolutionary processes in SLs."

In addition I think the coding process could be clarified a bit. "In coding morphological similarity, judgements must be made about non-discrete characters with potentially infinitely fine-grained differences." It is good that the authors are aware of this problem, but unclear how they decided to deal with it. There is for instance no information about who the coders were. I hope there were at least two coders and the similarity between the two raters was determined, but none of this is mentioned. And were the coders people who are signers? Whether some change in one of these parameters makes a meaningful distinction in a language differs between different sign languages. Prior exposure to a particular sign language may influence similarity judgements.

As a more general discussion point I wonder whether it matters that manual alphabets can be seen as a bridge between the sign language system and the sounds of the spoken language that surrounds it. To what extent is this analysis really fully independent of the surrounding spoken languages? Especially also since in part the analysis uses the sound of the spoken language as the base of comparison between characters. Perhaps this could be emphasized better.

The interpretation that is offered in the discussion and conclusion of the paper is talking about entire sign languages and relations between them, but there is a risk here of over-interpreting findings that are in the end only based on comparing forms of manual alphabets, along only two dimensions (handshape and movement). This can be seen as a bit of a stretch, since the alphabet

is just a very small part of a sign language, it is functioning in part as a separate system from the rest of the language (the authors themselves say: "an MA may be less integrated within a SL's lexicon, and MA forms or the MA as a whole, may in consequence be more readily subject to replacement compared with basic vocabulary.") and not even all sign languages have one. It may therefore make more sense to just talk about the conclusions that can be drawn from this work about the dispersion and spread of MA's instead of SL's.

So for instance the general statement:

"We have shown that computational phylogenetic methods can be applied to SL data to uncover new insights into the evolutionary histories of SLs, to generate new hypotheses about their relationships...", would be more correct as: "We have shown that computational phylogenetic methods can be applied to MA data to uncover new insights into the evolutionary histories of MAs, to generate new hypotheses about their relationships..." etc. And as such I think the paper really does what it promises and offers an interesting new contribution.

Finally one more general comment to be careful about big statements that are missing nuance: "The evolution of spoken languages is well understood" > I don't think many people who are working on this topic would agree...

Decision letter (RSOS-191100.R1)

02-Dec-2019

Dear Dr List,

On behalf of the Editors, I am pleased to inform you that your Manuscript RSOS-191100.R1 entitled "Evolutionary dynamics in the dispersal of sign languages" has been accepted for publication in Royal Society Open Science subject to minor revision in accordance with the referee suggestions. Please find the referees' comments at the end of this email.

The reviewers and Subject Editor have recommended publication, but also suggest some minor revisions to your manuscript. Therefore, I invite you to respond to the comments and revise your manuscript.

- Ethics statement

- Data accessibility

<http://datadryad.org/submit?journalID=RSOS&manu=RSOS-191100.R1>

- **Competing interests**

- **Authors' contributions**

- **Acknowledgements**

- **Funding statement**

Because the schedule for publication is very tight, it is a condition of publication that you submit the revised version of your manuscript before 11-Dec-2019. Please note that the revision deadline will expire at 00.00am on this date. If you do not think you will be able to meet this date please let me know immediately.

1) A text file of the manuscript (tex, txt, rtf, docx or doc), references, tables (including captions) and figure captions. Do not upload a PDF as your "Main Document".

- 2) A separate electronic file of each figure (EPS or print-quality PDF preferred (either format should be produced directly from original creation package), or original software format)
- 3) Included a 100 word media summary of your paper when requested at submission. Please ensure you have entered correct contact details (email, institution and telephone) in your user account
- 4) Included the raw data to support the claims made in your paper. You can either include your data as electronic supplementary material or upload to a repository and include the relevant doi within your manuscript
- 5) All supplementary materials accompanying an accepted article will be treated as in their final form. Note that the Royal Society will neither edit nor typeset supplementary material and it will be hosted as provided. Please ensure that the supplementary material includes the paper details where possible (authors, article title, journal name).

Kind regards,
Lianne Parkhouse
Editorial Coordinator
Royal Society Open Science
openscience@royalsociety.org

on behalf of the Associate Editor, and Professor Kevin Padian (Subject Editor)
openscience@royalsociety.org

Reviewer comments to Author:

Reviewer: 1
Comments to the Author(s)

The manuscript has been improved notably and it is a lot clearer now. I would, however, still have some minor comments which could help to improve the manuscript.

p4 r53-p5 r9. This paragraph is still quite unclear and would benefit from restructuring. First, it would help to say in the end of the first sentence that what is the point of the paragraph. Now the reason (that historically relevant information can be obtained from MA's) is said only in the end of the paragraph. Second, now when the first sentence mentions differences between historical comparative study of vocabulary and study focusing on MA's, the reader assumes that what follows is comparison of these two data types. In the current version, however, only peculiarities of MA system are listed without the comparison with basic vocabulary data. Also, it remained a bit unclear what does it mean in practice that SLs vary with regard to how integral the MA is to the language as a whole so it would be good to give an example of what kind of MAs are more integrated within a SL's lexicon and which are less integrated.

p7 r14. I couldn't find a reference to the SI section 4.1. This should be added to the main text.

p7 r38. Parentheses are missing from the end of (see Figures 5 and 6A-D in SI 4.3).

p8 r51 It is unclear to what the value $BS > 48$ is based on. I couldn't find the value 48 from Table 1.

p10 r12-13. Explanation of how different optimality criteria differ from each other could be removed to methods section.

p10 r17 For clarity, it would be good to refer to the specific SI Figure in which the 3 dimensional boxes concerning the Russian group are shown.

p10 r34-36. As the BS-values were quite low for many of the major groups, including Austrian-origin and French-origin (at least in Figure 3), I find it quite bold to say that each major group goes back to an independent founding event. It should be elaborated how the independent founding events are inferred here. Furthermore, the latter part of the sentence could be clearer if verbs would be added there.

p12 r52-53. "-- we argue that communities in Austria, France and Spain independently formed MAs using the early sources." Does this mean Austria, France and Spain SL communities formed MAs independently from each other or independently from early sources. With current formatting there is a chance of both kind of interpretations and it should be clarified what is meant here.

p12 r53-54. "We suggest that this independent formation supports identifying three separate SL lineages." In this sentence the chain of reasoning seems to be upside down as one would assume that the independent formation would be inferred from the three separate SL lineages and not vice versa. Furthermore, I could not find bootstrap values to the time-/taxon filtered NNets even though a lot of interpretation is based on the groups seen there (especially in the oldest time layer).

p13 15-17. The sentence "Preserved examples in books--" is unclear and should be rephrased.

p13 r21. "-- did not evolve directly from a shared Spanish-French origin." If the point here is to discuss whether French MA evolved from Spanish origin, shouldn't it be just "evolve directly from Spanish origin." It is thus unclear what is meant here with the Spanish-French origin.

Supplementary Information

p7 r1-2. "Handshapes for Latin <n> in contemporary American SL, Austrian SL and German SL were coded as similar and assigned ID 137." When checking what kind of handshapes these three languages have in the SI Figure 1, the version Austrian SL has looks notably different from American and German SLs. Thus it would be good to know whether this is correct and it is only matter of how to read those symbols.

Reviewer: 2

Comments to the Author(s)

This paper describes a model of Sign Language dispersion and relatedness on the basis of phylogenetic network methods. This approach is novel and has not been applied to the study of sign languages before, therefore I think there is value in sharing this work, as a first attempt to apply these methods, since others can build on and expand the approach described here. However I also think the paper in its current form has some issues that should be fixed before it is ready for publication. Some of the points I list below have to do with consistency issues that likely emerged between this version and the previous version of the paper. As I understand, a lot more prior knowledge on sign language dispersion from other sources such as historical data has been added to the background and as the first reviewer suggested, the focus of the modeling should be on filling some gaps in the (already quite detailed) picture we get from such other data,

instead of claiming that we know nothing and that comparing manual alphabets is going to change this. Some parts of the paper still resonate more with the second goal though.

For instance:

"scholars have been largely unsuccessful in grouping sign languages into monophyletic language families"

As some of the background discussed in the paper also suggests, we actually know quite a lot about how sign languages are related to each other through historical records and cultural information that is preserved since the origins of many currently established sign languages does not go as far back as the spoken languages we speak today.

And another contradiction: "While the exact origins of the MAs used in European educational institutions are unclear" and a bit further we read: "because the creation and transmission of many MAs are well-documented, the data provide a relatively well- understood test case for studying evolutionary processes in SLs."

In addition I think the coding process could be clarified a bit. "In coding morphological similarity, judgements must be made about non-discrete characters with potentially infinitely fine-grained differences." It is good that the authors are aware of this problem, but unclear how they decided to deal with it. There is for instance no information about who the coders were. I hope there were at least two coders and the similarity between the two raters was determined, but none of this is mentioned. And were the coders people who are signers? Whether some change in one of these parameters makes a meaningful distinction in a language differs between different sign languages. Prior exposure to a particular sign language may influence similarity judgements.

As a more general discussion point I wonder whether it matters that manual alphabets can be seen as a bridge between the sign language system and the sounds of the spoken language that surrounds it. To what extent is this analysis really fully independent of the surrounding spoken languages? Especially also since in part the analysis uses the sound of the spoken language as the base of comparison between characters. Perhaps this could be emphasized better.

The interpretation that is offered in the discussion and conclusion of the paper is talking about entire sign languages and relations between them, but there is a risk here of over-interpreting findings that are in the end only based on comparing forms of manual alphabets, along only two dimensions (handshape and movement). This can be seen as a bit of a stretch, since the alphabet is just a very small part of a sign language, it is functioning in part as a separate system from the rest of the language (the authors themselves say: "an MA may be less integrated within a SL's lexicon, and MA forms or the MA as a whole, may in consequence be more readily subject to replacement compared with basic vocabulary.") and not even all sign languages have one. It may therefore make more sense to just talk about the conclusions that can be drawn from this work about the dispersion and spread of MA's instead of SL's.

So for instance the general statement:

"We have shown that computational phylogenetic methods can be applied to SL data to uncover new insights into the evolutionary histories of SLs, to generate new hypotheses about their relationships...", would be more correct as: "We have shown that computational phylogenetic methods can be applied to MA data to uncover new insights into the evolutionary histories of MAs, to generate new hypotheses about their relationships..." etc. And as such I think the paper really does what it promises and offers an interesting new contribution.

Finally one more general comment to be careful about big statements that are missing nuance: "The evolution of spoken languages is well understood" > I don't think many people who are working on this topic would agree...

Author's Response to Decision Letter for (RSOS-191100.R1)

See Appendix B.

Decision letter (RSOS-191100.R2)

13-Dec-2019

Dear Dr List,

It is a pleasure to accept your manuscript entitled "Evolutionary dynamics in the dispersal of sign languages" in its current form for publication in Royal Society Open Science.

Kind regards,
Lianne Parkhouse
Editorial Coordinator
Royal Society Open Science
openscience@royalsociety.org

on behalf of the Associate Editor, and Professor Kevin Padian (Subject Editor)
openscience@royalsociety.org

Appendix A

Authors' response (“Evolutionary dynamics in the dispersal of sign languages”)

Our responses to all general and individual points raised by the editor and the reviewer are given below. To facilitate the navigation through this rather long response letter, we display comments by editor and reviewer in **blue font**, and slightly indented to the right, while our response is given in normal black font.

Comments by the editor

Unusually, and owing to the difficulty had in finding sufficient suitable reviewers, the Editors are opting to make their decision on the basis of one reviewer's report. The report offers a range of recommendations for improvement, and the referee will be invited to assess your revision, so we urge you to take due care in preparing the revision to ensure you fully address their concerns - please ensure you also include a full point-by-point response with your revision, and be aware that if you do not fully satisfy the reviewer that the paper is ready for publication, we may not be able to consider the paper further for publication. Good luck.

We would like to thank the editor for the effort taken in finding a suitable reviewer for our manuscript. We have now completed our revisions and addressed each of the reviewer's comments / questions, both in the manuscript and in the following point-by-point responses. Per the reviewer's main comments below, we have substantially re-organised the introduction and discussion sections of the main text, in particular, including more information about previously reported historical connections amongst sign languages (SLs). The discussion section is now also clearer, as previously reported information has been distinguished from our interpretations of the results. We have added additional tables and figures to the supplementary information (SI) in line with the reviewer's comments and where we judged that the information was better placed in SI instead of the main text. **Our modifications are all indicated by green font in the main document and the appendix.**

Comments by the reviewer

Reviewer 1, Comments to the Author(s): **The paper presents a novel study in which quantitative methods are used to resolve the relationships and dispersal of sign languages. The work is of scientific interest and in general well conducted. The paper would, however, benefit of a notable restructuring of text in the way that the earlier knowledge and hypotheses of the dispersal of sign languages would be presented thoroughly already in the introduction and not mainly in the discussion as it is now. This way it would be clearer what is known beforehand based on the earlier literature**

and what is found in this study, as in the current version this distinction is not always clear. Further and more specific comments are listed below.

We would like to thank the reviewer for numerous helpful comments and clarifying questions, which have certainly improved the manuscript. We have addressed each of the reviewer’s comments / questions in the main text, supplementary information (SI), and here in our point-by-point responses. In line with the reviewer’s comment about restructuring the text, we have substantially edited the introduction and discussion sections to distinguish the information that is based on previously-reported historical connections amongst sign languages (SLs) from information that relates to our interpretations of the results. In addition to the previously-reported historical connections surveyed in the introduction, we have included a table in SI (Section 2.3, Table 3) that gives a detailed overview of the languages, historical connections, and a selection of sources that have reported these connections. While this table may not be comprehensive, it synthesises, to the best of our knowledge, information in a range of sources and based on differing types of evidence. We have also attempted to make clear that, in many cases, this type of historical information is not available for a number of SLs, or that inferences about SL families have not always been based on language data.

In addition to the new table discussed above, we have added two figures to SI (Section 4.3, Figures 5 & 6A-D) with support values of splits corresponding to selected neighbourhoods mapped on the all-inclusive NNet (Figure 3 in the main text), and simplified versions of the C Nets. These additions are in lieu of the requested additional annotations of BS support values to the NNets in Figures 3 and 4 in the main text. We decided not to add these support values to the figures in the main text for the following reason. Because we established BS support under all three commonly used optimality criteria, adding all support values would create numerous additions to the NNets that we felt would make them incomprehensible visually. As mentioned above, instead of these annotations to Figures 3 and 4, we have provided the requested information in additional figures in SI, as well as explanations of the figures. We hope that this solution sufficiently addresses the reviewer’s request.

Revisions of the original manuscript are marked with dark green text to aid the reviewer in quickly locating revised / additional text, though this was not always possible for technical reasons (e.g., complete additional tables and figures); subtractions from the original manuscript are also not marked, though we have attempted to note these in the point-by-point responses below.

1. Introduction

p2 r 8-9: I find the first sentence of the introduction “Linguistic analyses of the world’s sign languages --” a bit confusing as now it sounds that it is the linguistic analyses that have shown that there can be a sign language that can be used as a mean of communication. However, as SLs have been used for communication for centuries, I would see that as a more important sign of the fact that SLs can be used in communication. Therefore, some changes should be made to the beginning of the first paragraph.

We have removed mention of “linguistic analyses”, as this was indeed confusing.

p2 r 11: It would be good to have some notions about the earlier/earliest existence of SL before going to the 18th and 19th century’s conventional and widespread SLs, especially as these older ones are also part of your dataset (Yebra and Bonet, are there others?).

We have added a brief discussion of the likely existence of SLs throughout human history, as well as references to research on so-called village signing communities and historical references to urban signing communities. In addition, we have provided a brief overview of the oldest records of manual alphabets (MAs) that scholars suggest are related to both one-handed and two-handed contemporary MAs. These changes can be found in the first paragraph of Section 1 in the revised version.

p2 r 20: Why there is a question mark after the reference 5?

This caused some confusion among us, as we did not, in fact, see a question mark in the compiled PDF we wrote. However, when checking the proofs produced by the journal’s system, we realized that this must have been introduced in the conversion process. The question mark should now have disappeared, and we also try to address other problems resulting from the post-processing of our file through the automatic conversion by the journal’s system.

p2 r 21-24: The spread of French SL to Netherlands, US and Brazil is mentioned in the text. It would be good if this kind of ‘from - to’ information would also be mentioned clearly about the other European SLs. It would serve as a proper introduction to the known spread and possible groups of SLs.

We have expanded the information about the dispersal of SLs in the revised introduction (main text, Section 1, paragraph 2) and have added a table (SI, 2.3) to the SI with more information and references for reported historical connections between the 40 contemporary SLs in the sample when known. In the revised introduction, we have also pointed out that while some connections between SLs have been reported in many sources, little has been published about the histories of many other SLs. In addition, scholars have often used differing types of evidence when reporting connections between SLs. We have attempted to summarise these issues in the introduction and provide more information in the SI. We hope this approach aligns with what the reviewer had in mind.

As the spread of French SL mentioned here matches with the lines spreading from France in Figure 5, it would be easier follow if this would be the case also with the other European SLs. Actually, if there is information about the possible routes of spread of SLs, these could be used as hypotheses for the analyses. This kind of re-structuring of the introduction would give it a more straightforward goal than the current “investigating evolutionary histories of SLs”.

As described above in our response to the previous comment, we have now revised the introduction to include the type of information about reported dispersals that the reviewer requested and have included a table in the SI with more detail and references. Figure 5 in the discussion section (Section 4) summarises information about hypothesised dispersals based on the results of our study. Therefore, Figure 5 differs from the information in the revised introduction and in the new table 2.3 (revised SI), which summarise previously-reported connections amongst SLs. These previous reports were not always based on language data, whereas our hypotheses are based on the results of the comparison of MAs. Because previously-reported historical connections amongst SLs are based on differing types of evidence—not always language data—and because there exist few higher-level classifications of the world’s SLs based on language data, an approach focusing on testing hypotheses about language relatedness amongst SLs did not seem feasible to us. These considerations explain our “investigatory” approach, in which we posit higher-order groupings of SLs based on language data.

In general: There is a mismatch with the reference numbers in the text and those in the reference list (e.g. I assume the references about recent work with phylogenetic methods should be 22 and 23 instead of 21 and 22 as it is now). It needs to be checked where this mismatch starts and the mismatch should be corrected.

This is, again, most likely due to the system’s compilation of our material. We made sure it is fixed by now.

There should be more general information in the introduction about sign languages and e.g. what are the ‘manual alphabets’ mentioned in row 43. There is more explanation about these in the materials and methods section (starting p3 r37), but I think the basics should be already in the introduction.

We have added a general description of MAs in the first paragraph of Section 1 in the revised version, as well as information about early 1- and 2-handed MAs.

Furthermore, there should be information about how many different SLs there is in the world so does the 76 manual alphabets cover all the SLs of the world or are they a sample of them? If those are a sample, how and why exactly those where chosen for the study?

We have added information from the Ethnologue (144) and Glottolog (193) about the total number of SLs in the world, as well as an explanation of the choice of SLs / manual alphabets, to the first paragraph of the Materials and Methods section (main text, Section 2) in the revised version. Our choice of SLs to include in the study was based mainly on the availability of quality data about the language’s MA that was also freely available online or via accessible print sources.

2. Materials and Methods

p2 r 57: Please explain in more detail what kind of a transcription HamNoSys does. Transcribes pictures of hand to different (sets of) symbols? Or something else?

We have added more detail about HamNoSys in the first paragraph of Materials and Methods (main text, Section 2) in the revised version by explaining which phonological parameters of the sign can be transcribed using HamNoSys symbols.

p2 r 58: Please explain in more detail how EDICTOR evaluates morphological similarity.

We have seen that our statement was not precise: EDICTOR does not judge similarity in any way, but rather allows experts to transparently reflect their similarity judgments about sequences. We have reworded this passage now, emphasising that EDICTOR specifically serves to avoid issues in the transparency of data coding, which is one of the major critiques regarding morphological analysis in biology, by making sure data is accessible both for humans and for machines.

p3 r58-p4 r10: It should be clarified what is the specific issue that has been the topic of debate concerning the linguistic status of MAs and their relationship to signed and written language. Do some SLs use MAs more than some others?

The original text may not have been clear about the issues at stake, and we have heavily revised the final paragraph of Section 2.1 to point out key considerations for an historical comparative study of MAs compared with a study of basic vocabulary. In the original text, we intended to highlight the fact that SLs vary with regard to how integral the MA is to the language as a whole, and that this variation differs compared to the integration of basic vocabulary in a language, which is assumed to be universally tightly integrated. Because of the variation found in SLs with respect to their MAs, there may be processes affecting the evolution of MAs that differ from processes affecting the evolution of basic vocabulary. We hope that our revisions now make these points clearer. These revisions are also relevant to the following question from the reviewer.

Concerning fingerspelling, I don't see what is the problem (or risk for losing the historical signal) if children learn the MAs before they are able to read. Or is the point that the MAs they learn as a kid are not the official ones?

We have revised the final paragraph of Section 2.1 to make clearer the relevance of children's acquisition of MA handshapes before learning to read. The aim of the text is to show that, although MAs and fingerspelling are tightly connected with written language, children acquire MA handshapes in some signing communities in a way that may parallel the acquisition of lexical vocabulary, ie, at early ages before learning to read. This situation is one end of the

“Evolutionary dynamics in the dispersal of sign languages”: Response to decision letter

spectrum along which signing communities / SLs may vary with respect to how integral MAs are to the particular SL and to how frequently fingerspelling is used. On the other side of the spectrum, we find signing communities / SLs that use fingerspelling much less frequently and in which MAs may not be as tightly integrated in the language. This difference may well influence the types of evolutionary processes that affect MAs, as we suggest in the revised version.

Now when it states “However, usage of fingerspelling varies cross-linguistically and across signers within the same signing community”, it does not yet tell whether they vary in frequency in usage or in the MAs they use. The paragraph should be clarified to answer these questions.

We have revised the final paragraph of Section 2.1 to make clear that it is the *frequency* of usage that we have in mind when discussing variation of usage.

2.2 Character coding, p4 r18: “We considered only handshapes and movements for determining similarity of MA forms.” In page 3 you say that an individual form of MA is comprised e.g. of a handshape with a particular spatial orientation. Is this spatial orientation taken into account when determining the similarity of MA forms or only the handshapes? This should be clarified in the text.

We took handshapes and movements, but not orientation or location, into account when coding similarity of MA forms. This is because many historical sources depict handshapes without reference to the body, making it difficult to determine what the actual spatial orientation of the handshape may have been. We have clarified this point in the main text of the revised version.

p4 r28: You write “In the example just mentioned, we compared handshapes representing Cyrillic B with Latin B and not V--“ However, in the example above it is not said that Cyrillic B is compared with Latin B but instead only mentioned that decision needs to be made whether the comparison is done with Latin B or V. Thus, please rephrase.

After re-reading our draft, we realised how easily our wording was leading to obvious misunderstandings. We have tried our best to reword this passage now, making clear that we see basically two possibilities for organising our comparison of MA forms that represent typologically different alphabets: form-based or sound-based. We opted for the first, due to evidence from historical records.

p4 r31: “When graphemic forms across two alphabet types are similar as in the case of Latin and Cyrillic just mentioned, the form of the grapheme typically becomes the basis of comparison.” In this the part “typically becomes” is quite vague. It should be specified in which cases the form of the grapheme was not used even those were similar.

“Evolutionary dynamics in the dispersal of sign languages”: Response to decision letter

We have revised this sentence and removed the word “typically” to make clear that we used the form of the grapheme as the basis of comparison when the forms were similar.

p4 r36-37: Explanation about character mapping and what can be achieved with it should be elaborated.

We are sorry for the brevity on character mapping. We have now substantially modified the according passages in the main text and supplement.

p4 r41: Is ‘concept’ the basic unit in your data (cf. ‘meaning’ in basic vocabulary lists)? If yes, the explanation about the example would be clearer if it would start by explaining that it is about “concept g” and only after that the divisions within it.

We have rearranged the explanation of the example in the second paragraph of Section 2.2, Character Coding. In the revised version, the concept is highlighted first, before explaining the various letters (the “narrow concepts”) represented in the comparison.

p4 r50-57: Was the similarity obtained automatically by EDICTOR tool or was it based on subjective human evaluation? Was it always clear which handshapes are assigned similar and which not? These issues should be elaborated in the text.

We have revised the first paragraph of Section 2.1 to make clear that EDICTOR was used only for storing the data set and our analyses, including similarity judgements, and not for automatic similarity judgements.

We have revised the first paragraph of Section 2.2 to explain the criteria for similarity judgements. In determining similarity, we considered specifications for finger extension, finger bending, extended finger separation, thumb configuration, and movement. If these were all the same, we coded the MA forms with the same ID. If any of the specifications were different, we coded the forms with different IDs. We have also explained in the revised version that judging morphological similarity is always challenging because it involves determining category boundaries across potentially infinitely fine-grained differences. This was especially the case in our judgements of the similarity of finger bending in handshapes. These challenges are one reason that we have made our data set and similarity judgements openly accessible to other researchers, a point that we reiterate in the revised version of Section 2.2.

2.3. Phylogenetic analysis

Please elaborate in this section which different sets of analyses with which data were done (whole data set, divisions into time layers etc.). What is the purpose of using both NNets and C Nets? This should be mentioned also in the main text and not only in the Supplementary Information. Please also refer to Supplementary Information for more

information and explain which results are shown in the Results section of the main text and which in the SI.

We added the two following sentences at the according places in the main text:

“NNets were inferred for the complete taxon set and time-filtered taxon subsets: all MAs from before 1840, later historical MAs including youngest pre-1840 MAs as reference points (potential sources), and contemporary and post-1950 MAs.”

“While NNets handle signal incompatibility well, they are restricted to two dimensions. In our case, an MA may derive from more than two sources and can show affinities to more than two unrelated groups of SLs. Such complex relationships may be captured in a bootstrap pseudoreplicate sample and visualised using CNETs. For instance, if an MA *xy* shares CogIDs with three different SL groups, the bootstrap replicate trees may reflect this situation by showing a 3-way split support for each topological alternative (*xy* is placed within a different group in each pseudoreplicate) and an according 3-dimensional box in the CNet.”

3. Results

p6_r29: It would be clearer if the third group (British) would be presented only after the Austrian and French groups are clarified and when starting to explain about the Afghan-Jordanian group.

We have reorganised the first paragraph of Section 3 according to the reviewer’s suggestion.

Figure 3: It would be good to have the bootstrap values of different groups also in the plot as some of the main groups are clearly supported while others are not.

The information about group support can be found in Table 1, a comprehensive list is included as a spreadsheet in lists.xlsx. We have decided not to add BS support values to Figure 3 for the following reasons. Figure 3 (and Fig. 4, see below) already includes a large amount of information. Because we established BS support under all three commonly used optimality criteria, adding support values is not feasible. Instead, we added two figures to the supplement: support values of splits corresponding with selected neighbourhoods mapped on the all-inclusive NNet, and simplified versions of the CNETs (colour-annotated full versions in Splits-NEXUS-format are included in the supplementary material). Note that the NNets in Fig. 4 are based on the same data basis compared with the all-inclusive NNet shown in Fig. 3.

Also, in Figure 3, some SLs are marked with a colored circle that differs from the time scale given in the figure: Russian 1835 is red, French 1800 is blue, Yebra 1593 is purple and Austrian 1786 is green. The reason for the differing coloration of these SLs should be mentioned in the Figure legend or changed so that the coloring follows the given time slots.

“Evolutionary dynamics in the dispersal of sign languages”: Response to decision letter

We have now changed all circles into the appropriate shade of grey in line with the figure legend. The coloured circles referred to the oldest MAs in each group (potential sources).

p8_r31: It should be mentioned in the materials and methods that NNets are done for different time periods separately. It should also be said why some historical samples were excluded from the analyses as now these are just mentioned in the end of the results section.

We added an according sentence to Section 2, M&M (see above).

We have revised the last sentence in Section 3, Results. There were no historical samples excluded from the analyses. The intention of the sentence is that the data set does not include any historical examples of SLs from the British-origin and Afghan-Jordanian groups, and that this explains their absence from Figure 4, which focuses on MAs with historical examples.

p8_r32: The time slots are different in the text and in the legend of Figure 4. They should be in line with each other.

We have now corrected the main text and legend of Figure 4 in Section 3 of the revised version to bring the time periods in line with each other.

p8_r37: Please add with a reference when did the French SL MA disperse to the Americas?

This comment and the one immediately following refer to statements that were intended as interpretations of our results, and not as background information that we might be able to reference. In line with the reviewer’s other comments, namely, restructuring the paper to include more background information of known historical connections amongst SLs in the introduction, we have included references to the spread of French SL to the Americas in that part of the revised version. In addition, we have included a phrase in the relevant paragraph of Section 3 to directly address this comment and to make clearer that statements about the dispersal of the French MA and the homogenising effect of the International MA are interpretations of the results. We hope that this addition makes the text clearer and thank the reviewer for the helpful clarification.

p8_r53-59: Explanation about International Sign and its homogenizing effect on European SLs (with references) should be moved to the introduction.

Please see our response to the previous comment. The statement about the homogenising effect of the International MA was intended as an interpretation of the results, and we hope that this is now clearer in the text. In addition, we have added a sentence to the revised introduction

“Evolutionary dynamics in the dispersal of sign languages”: Response to decision letter

(final sentence of paragraph 3) to provide background information about the International MA with references.

Figure 4: It would be good to have bootstrap values also in Figure 4, at least for the edge bundles of the main groups. In addition, in the bottom NNet the split connecting French group with German 1820, Hungary 1827 and Austrian 1786 looks longer than then one splitting only the French group. Would there be something interesting to discuss about? It should also be noted what the dashed lines denote.

Regarding the addition of BS support values to Figure 4, please see our response above to the reviewer’s similar comment about Figure 3. We have added a brief statement clarifying that the dashed lines link languages in multiple NNets.

Figure 5: The two greens and the two shades of blue look very similar in the Figure. It would be good to give a bit more different colours to these.

We have made modifications to Figure 5 by adding white and black outlines to the arrows. We have decided not to change the colours themselves because the colours are used across the preceding figures (and throughout the supplement): (mid-dark) green for Austrian-group, turquoise for French-group, light and dark blue for Swedish and English groups. The visual problem here may be caused by the background colours (it may be especially challenging for those suffering from some form of yellow-blue blindness). We think that our modifications make the arrows stand out against the background better. We thank the reviewer for pointing this out.

In the legend it says “Timeline reflects approximate years of first transmission of SLs, typically coinciding with establishment of schools.” To what is the first transmission time of SL based on or is it the time of the establishment of the school?

We have removed the word “typically” from Figure 5’s legend in the revised version. The legend now states that first transmission of SLs is estimated based on the establishment of schools for the deaf, covering the majority of cases—the original intention of the word “typically”—or of known migrations of signers, as for example in the migration of British signers to Australia before the establishment of a school for the deaf there.

4. Discussion

p10 r57-58: The first sentence in Discussion is an example of a sentence of which I am not completely sure what it is based on. If it is based on the time layer analysis and Figure 4, it is problematic as those analyses did not include British and Swedish group, which nevertheless are part of Figure 5.

We have added a sentence to the first paragraph of Section 4 in the revised version to make clear that the dispersal pathways in Figure 5 are based on a synthesis of our comparison of

“Evolutionary dynamics in the dispersal of sign languages”: Response to decision letter

MAs, as well as historical information about the establishment of schools for the deaf, migrations of signers, and publication of the pre-18th century MAs. In this way, Figure 5 interprets the study’s results in light of other historical, extra-linguistic evidence.

p11 r19: “While the French MA was clearly formed using Spanish sources--.” It is again unclear whether this is based on earlier literature or the results of this study. Please make this division clear here and also in other parts of the text.

We have revised the first sentence in paragraph 2 of Section 4 to clarify that previous reports indicate that Spanish sources were known in France and were likely used in forming the French MA. We then argue that the results of the study indicate that the French MA was formed independently, despite the historical connection between the two countries.

p11 r30: “--in which just eleven forms remained unmodified--” Eleven forms out of how many?

We have added a phrase to paragraph 2 of Section 4 in the revised version to show that 11 out of 22 forms remained unmodified.

p11 r31: “In adapting the early Spanish sources, the signing community in Paris changed them substantially ... , which resulted only minimal modifications to the existing forms.” It is now a bit unclear whether there were substantial or minimal modifications to the early Spanish sources. This should be clarified.

We have revised the first paragraph of Section 4, adding information about the numbers of unmodified MA forms and describing slight modifications to other forms. In addition, we have added a figure and description to SI 4.4, providing a comprehensive look at the earliest (Yebra and Bonet) MAs and how they evolved in the earliest examples in our sample of Spanish (1815), French (1800), and Austrian (1786) MAs. In addition, we have removed the final phrase in the sentence quoted by the reviewer; in particular, the phrase “which resulted in only minimal modifications to existing forms” has been removed because it is unnecessary for making the relevant point.

p11 r33-36: Could the sentence starting “It is clear that the--” be said in a more straightforward way as the current version is quite complex.

We have revised this sentence, breaking it into two sentences and moving the revised sentences earlier in the paragraph for clarity.

p11 r37-41: Example of a sentence that should be already in the introduction and not only in the discussion. The following sentence “Perhaps because --” would also make an appropriate hypothesis what to study and should also be mentioned already in the introduction.

“Evolutionary dynamics in the dispersal of sign languages”: Response to decision letter

We have revised the introduction to include this sentence, as well as other information on the major lineages outlined in the results section.

p12 r45-55: It should be mentioned already in the introduction that there is also a 2 handed MA and that for this reason the British group may be expected to appear as a group of its own.

We have included in the revised introduction a brief discussion of the dispersal of British SL to Australia and New Zealand in the 19th century and cited scholars who have claimed that the languages are related. We have also noted that the British MA is 2-handed.

p12 r46: Reference is denoted only by a question mark.

See our statement above. In this version, we have made sure that the proofs actually look the same as the paper we compiled on our computers. We apologize for not doing so earlier and learned our lesson for the future.

In general: It would be good to discuss shortly the possible role of iconicity to the results as perhaps some MAs are more iconic and thus less likely to change.

We have included a discussion of iconicity in Section 4 of the revised version, at the end of the paragraph about the Russian lineage. Our discussion incorporates the reviewer’s comment and also identifies a potential tendency for MAs representing typologically similar alphabets to converge on handshape forms independently. As the reviewer suggests, iconicity may indeed affect the interpretation of our results, and our additions make this clearer.

5. Conclusion

It would be good to summarise in conclusions what were the main findings and how they differed from what was expected.

We have now added sentences summarising our findings to the revised conclusion.

p13 r31-32: Is there a word missing in “--we suggest that our analysis [could/should] be taken in future research --“?

This sentence is grammatical in one of the authors varieties of native English. In case this clashes with the journal’s preferred English variety, or leads to too much confusion, we will gladly modify it. For now, we have left the sentence as is, and would kindly ask the editor and the reviewer to confirm the journal’s preference.

Supplementary Information

1. Organisation of the supplementary material

The first link given in the text does not work.

Sorry, the link has now been updated. It will lead directly to the EDICTOR application.

2.2 Historical manual alphabets

The word ‘in’ is twice one after another in the sentence.

We have done so.

3.2 Coding

Apparently, the year of the historical Estonian SL is missing both from the table and from the text, or I do not see how the concept of h could be both the narrow concept of h and n in contemporary Estonian.

In fact, the year is not missing from the table or the text. Both narrow concepts (h and n) are covered by the concept h in the way that we have treated this type of cross-alphabet comparison. We have included more explanation for why this is so in the final paragraph of Section 3.2 in SI. In brief, because of our decision to homologise the Estonian SL form representing n with the Russian SL form representing н, and because Estonian SL also has a form representing h, there are two forms from Estonian SL in the h-comparison.

4.2 Character mapping

Please elaborate how the character mapping was done and rephrase the section and subsections to make it more like the main text, e.g. rephrase the text in the way that it starts with how the median networks were reconstructed and not with reference to figures.

We added a detailed general description of the mapping process at the start of the paragraph, introducing “lingotypes” to address the unique binary sequences encoding the different handshapes of the mapped concepts (standard latin letters).

Also, add legends to the figures in sections 4.2.1, 4.2.2 and 4.2.3 and give these figures appropriate figure numbers which you refer in the SI text.

Yes, a very good point we hadn’t thought of before. We have now numbered all figures, and also reference them in the relevant passages of the main text, where we refer to the SI. We have done the same now also for all tables, as it turned out to be much clearer to reference them in this way.

“Evolutionary dynamics in the dispersal of sign languages”: Response to decision letter

More specifically, in page 7, row 4 the replacement of Dutch version by Austrian is denoted as (“At->Du”) whereas in the figure it is other way around. This should be changed so that it is correct in both.

The text has been corrected.

4.2.1. It is not immediately clear to which the “list” refers to. This should be made clearer.

We added an according subheading and table heading (the list is now Table 4, figures are labelled consecutively, and referenced accordingly) and re-phrased the introductory sentence.

Authors' 2nd response (“Evolutionary dynamics in the dispersal of sign languages”)

We express our gratitude to the two reviewers and the editor for taking the time to help us improve our manuscript. Our responses to all general and individual points raised by the Editor, Reviewer 1, and Reviewer 2 are given below. To facilitate the navigation through this response letter, we display comments by editor and reviewers in **blue font**, and slightly indented to the right, while our responses are given in black font.

Editorial remarks

We have now updated the text and added those formally required sections that were missing before. In addition to the data availability statement, we leave a statement on supplementary information and material, as we consider it important that these materials are easy to find for colleagues, and we know from experience that the first way to search a document is to search for keywords on supplementary information or supplementary material. Note that we host our data on Zenodo, from where they are now also often automatically included in FigShare. We insist on Zenodo as a first host due to the specific integration of this repository with other services we use; but if required by the journal, we are happy to provide a clone for FigShare.

Comments by Reviewer 1:

Reviewer 1, Comments to the Author(s): **The manuscript has been improved notably and it is a lot clearer now. I would, however, still have some minor comments which could help to improve the manuscript.**

We would like to thank this reviewer, again, for many helpful comments. We have addressed each of the reviewer's comments in the main text, supplementary information (SI), and here in our point-by-point responses.

p4 r53-p5 r9. **This paragraph is still quite unclear and would benefit from restructuring. First, it would help to say in the end of the first sentence that what is the point of the paragraph. Now the reason (that historically relevant information can be obtained from MA's) is said only in the end of the paragraph. Second, now when the first sentence mentions differences between historical comparative study of vocabulary and study focusing on MA's, the reader assumes that what follows is comparison of these two data types. In the current version, however, only peculiarities of MA system are listed without the comparison with basic vocabulary data. Also, it remained a bit unclear what does it mean in practice that SLs vary with regard to how integral the MA is to the language as a**

whole so it would be good to give an example of what kind of MAs are more integrated within a SL's lexicon and which are less integrated.

We have revised this paragraph further clarifying the point of the paragraph in a revised first sentence, per the reviewer's suggestion. We have made the comparison with basic vocabulary more explicit, highlighting how sampling from MAs instead of from basic vocabulary may affect interpretations of the results.

p7 r14. I couldn't find a reference to the SI section 4.1. This should be added to the main text.

A reference to SI 4.1 has been added to section 2.3 in the main text.

p7 r38. Parentheses are missing from the end of (see Figures 5 and 6A-D in SI 4.3).

We have added the missing closing parenthesis.

p8 r51 It is unclear to what the value $BS > 48$ is based on. I couldn't find the value 48 from Table 1.

We have added the missing reference. Sheet *Support in lists.xlsx* in the Supplementary Material includes all information about alternative BS supports visualised in Supplementary Information 4.3.

p10 r12-13. Explanation of how different optimality criteria differ from each other could be removed to methods section.

Since such data have never been analysed in this way (to our knowledge), we would prefer to keep this explanation in the Results section.

p10 r17 For clarity, it would be good to refer to the specific SI Figure in which the 3 dimensional boxes concerning the Russian group are shown.

References to figures have been added.

p10 r34-36. As the BS-values were quite low for many of the major groups, including Austrian-origin and French-origin (at least in Figure 3), I find it quite bold to say that each major group goes back to an independent founding event. It should be elaborated how the independent founding events are inferred here. Furthermore, the latter part of the sentence could be clearer if verbs would be added there.

The three points mentioned in the latter part of the sentence are evidence supporting the inference of independent founding events. The first point - "no to little support for the deepest

splits” - refers to the result on the bottom of page 8, where we indicate that all other alternatives (i.e., any possible alternative) are unsupported. Thus, the most straightforward inference based on this result is that the eight groups evolved independently. If some of them would share a single founding event, the lineage(s) originating from this (these) founding event(s) should find some super-group BS support.

We have rephrased this sentence to better convey the connection between evidence and inference, and have also rephrased the latter part of the sentence for clarity.

p12 r52-53. “-- we argue that communities in Austria, France and Spain independently formed MAs using the early sources.” Does this mean Austria, France and Spain SL communities formed MAs independently from each other or independently from early sources. With current formatting there is a chance of both kind of interpretations and it should be clarified what is meant here.

We have rephrased this sentence to clarify that we intend the first interpretation, namely, that the communities used the same sources, but formed their MAs independently of one another.

p12 r53-54. “We suggest that this independent formation supports identifying three separate SL lineages.” In this sentence the chain of reasoning seems to be upside down as one would assume that the independent formation would be inferred from the three separate SL lineages and not vice versa.

This sentence was indeed not precisely what we intended. We have adjusted the sentence to better reflect the intended interpretation, namely, that inferring independent formation events is ipso facto an inference of separate lineages.

p12 r53-54. Furthermore, I could not find bootstrap values to the time-/taxon filtered NNets even though a lot of interpretation is based on the groups seen there (especially in the oldest time layer).

As pointed out in our first-round response (see following quote), adding BS values would overload the in-text graphics.

“We decided not to add these support values to the figures in the main text for the following reason. Because we established BS support under all three commonly used optimality criteria, adding all support values would create numerous additions to the NNets that we felt would make them incomprehensible visually. As mentioned above, instead of these annotations to Figures 3 and 4, we have provided the requested information in additional figures in SI, as well as explanations of the figures. We hope that this solution sufficiently addresses the reviewer’s request.”

p13 15-17. The sentence “Preserved examples in books--“ is unclear and should be rephrased.

We have now rephrased this sentence. Our intention is to offer a possible explanation for the stability (i.e., lack of change) in most MA forms between Bonet's MA in 1620 and the Spanish MA from 1815. The possible explanation is that Bonet's MA was not used widely in a signing community in Spain until the early 19th century, and so may not have been subject to evolutionary processes connected with usage in a community.

p13 r21. "-- did not evolve directly from a shared Spanish-French origin." If the point here is to discuss whether French MA evolved from Spanish origin, shouldn't it be just "evolve directly from Spanish origin." It is thus unclear what is meant here with the Spanish-French origin.

The purpose of using "shared Spanish-French" origin instead of the suggested "Spanish" origin is to connect with our argument of independent formation of the Spanish and French MAs. Using the suggested "Spanish" origin may confuse readers, since we are arguing that the French MA drew on Yebra and Bonet, which are both Spanish sources in the sense that they came from Spain; but that the origin of the French MA is independent of the MA that evolved later in Spain once a signing community formed there.

Supplementary Information

p7 r1-2. "Handshapes for Latin <n> in contemporary American SL, Austrian SL and German SL were coded as similar and assigned ID 137." When checking what kind of handshapes these three languages have in the SI Figure 1, the version Austrian SL has looks notably different from American and German SLs. Thus it would be good to know whether this is correct and it is only matter of how to read those symbols.

Indeed, the difference between the symbols is much larger than the small difference between the handshapes, which mainly comes down to the position of the thumb. In the American and German SL handshapes, the thumb rests on the proximal phalangeal bone of the 4th finger and thus protrudes somewhat between the third and fourth fingers; whereas in the Austrian SL handshape, the thumb rests slightly farther forward on the proximal interphalangeal joint of the fourth finger, and thus does not protrude between the third and fourth fingers, though it is still visible. The HamNoSys symbol used for the American and German SL handshapes highlights this protrusion of the thumb, whereas the symbol for the Austrian SL handshape shows that the thumb is opposed but not protruding. We have included a sentence noting this difference at the appropriate point in SI 3.2.

Comments by Reviewer 2:

Reviewer 1, Comments to the Author(s): This paper describes a model of Sign Language dispersion and relatedness on the basis of phylogenetic network methods. This approach is novel and has not been applied to the study of sign languages before, therefore I think there is value in sharing this work, as a first attempt to apply these

methods, since others can build on and expand the approach described here. However I also think the paper in its current form has some issues that should be fixed before it is ready for publication. Some of the points I list below have to do with consistency issues that likely emerged between this version and the previous version of the paper. As I understand, a lot more prior knowledge on sign language dispersion from other sources such as historical data has been added to the background and as the first reviewer suggested, the focus of the modeling should be on filling some gaps in the (already quite detailed) picture we get from such other data, instead of claiming that we know nothing and that comparing manual alphabets is going to change this. Some parts of the paper still resonate more with the second goal though.

We would like to thank this reviewer for numerous helpful comments and clarifying questions, which have certainly improved the manuscript. In line with the reviewer's comments about consistency, we have clarified our descriptions of known background information, in particular, distinguishing between what is known about the historical contexts within which MAs and SLs have evolved; and the relative lack of knowledge about the relatedness of the languages themselves based on linguistic evidence. We have also clarified our approaches to coding and interpreting the results of the study.

For instance:

"scholars have been largely unsuccessful in grouping sign languages into monophyletic language families"

As some of the background discussed in the paper also suggests, we actually know quite a lot about how sign languages are related to each other through historical records and cultural information that is preserved since the origins of many currently established sign languages does not go as far back as the spoken languages we speak today.

The reviewer rightly points out that there is ample documentary evidence for some international connections among educators of the deaf, educational institutions, and sometimes even Deaf signers. In response to Reviewer 1's comments during the first round of revisions, we already added an overview of much of this information in the introduction and provided a table in SI, Table S2.3, with an extensive overview of previously reported connections among sign languages. However, while it is true that we have a good understanding of the contexts in which some sign languages may have evolved, the relatedness of the languages themselves is a different question that cannot be answered satisfactorily without looking at the linguistic evidence. We have added the phrase "using traditional methods" to the end of the sentence in question to make this point clearer.

And another contradiction: "While the exact origins of the MAs used in European educational institutions are unclear" and a bit further we read: "because the creation and transmission of many MAs are well-documented, the data provide a relatively well-understood test case for studying evolutionary processes in SLs."

This does indeed appear contradictory on first reading. However, we feel the apparent contradiction is mainly due to the phrases being juxtaposed in the reviewer's comment, whereas in the main text, they are in separate contexts. The point of the first sentence in the first paragraph of the introduction is to highlight that scholars are unsure about the ultimate origins of the one-handed MAs that we find in Yebra and Bonet. There are some indications that these MAs may have developed from other MAs used in monasteries (Plann 1997, Padden & Gunsauls 2003), but the evidence is scant and ambiguous.

The point of the second sentence, which comes in the second paragraph of section 2.1, is to contrast the documentary evidence that we have for many MAs—for example, in the form of drawings from the 18th and 19th centuries—with the relative lack of evidence we have about the lexical signs that were used for basic vocabulary in these historical communities.

However, we do recognize the potential for confusion that the reviewer points out, and we have now revised the second sentence in section 2.1 and restructured the paragraph slightly: "First, many of the historical contexts in which MAs were created and transmitted have been well-documented, as have the MAs used in these communities. There exist far fewer historical dictionaries of the world's SLs compared with historical examples of MAs, though some do exist. Thus, the data provide a relatively well-understood test case for studying evolutionary processes in SLs."

In addition I think the coding process could be clarified a bit. "In coding morphological similarity, judgements must be made about non-discrete characters with potentially infinitely fine-grained differences." It is good that the authors are aware of this problem, but unclear how they decided to deal with it. There is for instance no information about who the coders were. I hope there were at least two coders and the similarity between the two raters was determined, but none of this is mentioned. And were the coders people who are signers? Whether some change in one of these parameters makes a meaningful distinction in a language differs between different sign languages. Prior exposure to a particular sign language may influence similarity judgements.

The reviewer is right in that this information was not explicitly mentioned in the text (only in the author contributions). We have addressed the question now by adding two clear statements on the coding in the main text.

As a more general discussion point I wonder whether it matters that manual alphabets can be seen as a bridge between the sign language system and the sounds of the spoken language that surrounds it. To what extent is this analysis really fully independent of the surrounding spoken languages? Especially also since in part the analysis uses the sound of the spoken language as the base of comparison between characters. Perhaps this could be emphasized better.

The reviewer raises an interesting point about the connection between the MA and the surrounding spoken, or indeed written, language. In one sense, it seems unlikely that any analysis of a language or of a subsystem within a language could be fully independent of the community's social or linguistic contexts. For this study, there may be a fruitful analogy here to processes such as selection affecting the evolution of organisms. For example, MAs representing Cyrillic-based alphabets are grouped together in this analysis. It seems clear based on the results that this grouping is due mainly to vertical evolutionary processes, but there is also the possibility that the script itself, i.e., an aspect of the context in which the MAs evolved, has led to convergent evolution. This is an important point that we have highlighted in our discussion of tree-incompatible signal in section 2.3, and in the SI in section 4.4.

The interpretation that is offered in the discussion and conclusion of the paper is talking about entire sign languages and relations between them, but there is a risk here of over-interpreting findings that are in the end only based on comparing forms of manual alphabets, along only two dimensions (handshape and movement). This can be seen as a bit of a stretch, since the alphabet is just a very small part of a sign language, it is functioning in part as a separate system from the rest of the language (the authors themselves say: "an MA may be less integrated within a SL's lexicon, and MA forms or the MA as a whole, may in consequence be more readily subject to replacement compared with basic vocabulary.") and not even all sign languages have one. It may therefore make more sense to just talk about the conclusions that can be drawn from this work about the dispersion and spread of MA's instead of SL's.

So for instance the general statement:

"We have shown that computational phylogenetic methods can be applied to SL data to uncover new insights into the evolutionary histories of SLs, to generate new hypotheses about their relationships...", would be more correct as: "We have shown that computational phylogenetic methods can be applied to MA data to uncover new insights into the evolutionary histories of MAs, to generate new hypotheses about their relationships..." etc. And as such I think the paper really does what it promises and offers an interesting new contribution.

On analogy to phylogenetic studies in evolutionary biology, we view MAs as "body parts" or "genes" of SLs, the latter representing the whole organism. Also in computational linguistics of spoken languages, studies typically sample basic vocabulary and infer relationships based only on that sample. Thus, inferred relationships of MAs will tell us *something* about relationships of SLs as a whole. As the reviewer has pointed out, we have already included statements about how sampling from MAs could affect the interpretation of our results, and the revised final paragraph of section 2.1 in the main text also emphasises this point. We have now ensured that this is clear through the entire text. For example, we have rephrased the sentence highlighted by the reviewer as follows:

"We have shown that computational phylogenetic methods can be applied to MA data to uncover new insights into the evolutionary histories of SLs, to generate new hypotheses about their relationships..."

Finally one more general comment to be careful about big statements that are missing nuance: "The evolution of spoken languages is well understood" > I don't think many people who are working on this topic would agree...

Indeed, this statement lacks nuance. We have now revised the beginning of the abstract.